# 3D MeanFlow: One-Step Point Cloud Completion and Generation via Average-Velocity Transport

Haowen Zhong [1]  Jiujun Cheng [1]  Haowen Wang [2]  Chao Wei [1]  Lu Yang [1]  Qichao Mao [1]  Shangce Gao [3]

## Abstract

Point cloud completion and generation are important across many 3D tasks, where both fidelity and sampling efficiency matter. Prevailing high-fidelity approaches rely on long sampling schedules, which incur substantial inference latency. Few-step alternatives typically use rectification or distillation, leading to multi-stage training pipelines and potential quality trade-offs. We present 3D MeanFlow (3DMF), a distillation-free model that performs one-step average-velocity transport for point cloud completion and generation. We optimize an instantaneous-average consistency objective and impose a shape-level constraint to stabilize training. Additionally, we introduce PointPlug, integrating completion into 3D object detectors and evaluating its impact. PointPlug uses adaptive selection that balances benefit and latency. Across standard benchmarks, 3DMF achieves one-step sampling with an order-of-magnitude speedup while maintaining competitive fidelity. On nuScenes and KITTI, inserting PointPlug improves all evaluated detectors under comparable settings.

## 1. Introduction

Point cloud completion and generation support a wide range of applications in autonomous driving (Li et al., 2022; Huang et al., 2022b), robotics (Ni et al., 2020; Huang et al., 2022a), and virtual reality (Kharroubi et al., 2019). Completion restores geometry lost to occlusion and sensor sparsity and directly impacts downstream detection, tracking, and

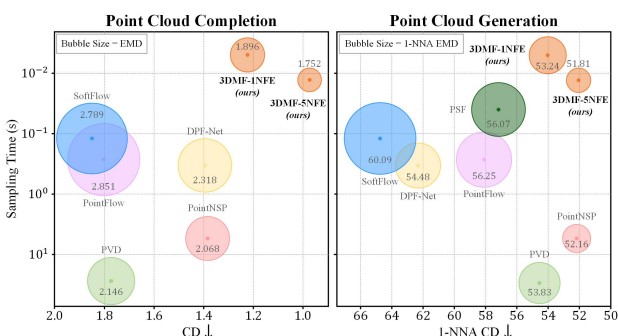

*Figure 1.* On the ShapeNet *Car* category, 3DMF delivers competitive fidelity for completion and generation. 3DMF-1NFE samples in about 5 ms per shape, while 3DMF-5NFE further improves accuracy with modest additional latency.

mapping, especially at long ranges. Generation provides diverse, controllable scenes for simulation and data pipelines, improving coverage of rare cases while reducing annotation cost. In operational settings, these models must satisfy real-time latency constraints while sustaining high-throughput data production (Wang et al., 2024). Thus, two properties are critical and non-negotiable: output quality, which governs downstream accuracy and safety, and sampling efficiency, which governs latency, throughput, and energy.

Most recent work (Vahdat et al., 2022; Zhu et al., 2025; Ren et al., 2024; Luo & Hu, 2021; Zhou et al., 2021) on point cloud completion and generation is built on multi-step diffusion. Allocating many sampling steps to approximate the generative trajectory often improves geometric accuracy and structural coherence, but the inference cost and latency grow roughly linearly with the step count, straining real-time budgets. To reduce this cost, few-step generation methods (Liu et al., 2023; Wu et al., 2023b; Zhu et al., 2024) compress or distill the long sampling chain into a handful of forward passes, sometimes a single pass, substantially lowering latency. Even with rectification (Liu et al., 2023; Lipman et al., 2023; Albergo & Vanden-Eijnden, 2023) or consistency techniques (Song et al., 2023; Song & Dhariwar, 2024; Geng et al., 2025b; Kim et al., 2024) to mitigate approximation error, quality degradation is common and training pipelines become more complex. Across practical settings, this progression still fails to deliver a satisfactory

---

[1]School of Computer Science and Technology and Key Laboratory of Embedded System and Service Computing (Ministry of Education), Tongji University, Shanghai, China [2]School of Computer Science and Technology, Anhui University, Hefei, China [3]Faculty of Engineering, University of Toyama, Toyama, Japan. Correspondence to: Jiujun Cheng <chengjj@tongji.edu.cn>, Haowen Wang <wanghaowen@ahu.edu.cn>.

quality-speed trade-off.

Recent advances in few-step generation have shown strong progress (Geng et al., 2025a; Frans et al., 2025; Wu et al., 2025). MeanFlow (Geng et al., 2025a) introduces a distillation-free training paradigm that learns an average-velocity field and ties it to the instantaneous velocity through an identity, enabling one-step sampling while maintaining fidelity. In light of this modality-agnostic, one-step formulation, we start from the following 3D modeling assumption: the generative state is a finite set of 3D point coordinates, and the transport is a learned per-point displacement field whose one-step application advances the entire shape.

However, applying this per-point displacement view directly to 3D point clouds poses two challenges. First, point clouds are unordered sets with variable cardinalities. If the average-velocity identity or its targets are order sensitive, updates become order-dependent, which may lead to inconsistent one-step directions and unstable behavior. Second, 3D tasks are highly sensitive to global shape and topological consistency. When training is driven primarily by local, neighborhood-level velocity targets and lacks a shape-level constraint, local errors remain unregularized across parts and scales. These errors accumulate and induce drift in the overall shape and proportions.

To address these challenges, we propose 3D MeanFlow (3DMF), a teacher-free, one-step model for 3D point cloud completion and generation. First, we restate the average-velocity identity in a set-based form so that learning remains permutation-equivariant and insensitive to point ordering or sampling patterns. Second, we introduce a shape-level consistency term that imposes a global constraint and prevents drift in the overall shape and proportions. Building on these designs, 3DMF learns an average-velocity field under a consistency objective. This realizes an average-velocity transport over point sets, aligning average and instantaneous velocities and enabling a single stable update. Across standard 3D completion and generation benchmarks, 3DMF achieves competitive fidelity while sampling more than $8\times$ faster than a representative published one-step baseline, PSF (Wu et al., 2023c), under comparable settings. Additionally, we present PointPlug, a plug-in module that adds one-step completion to modern 3D detectors by adaptively selecting regions of interest and feeding the completed geometry back to update boxes and scores. On nuScenes (Caesar et al., 2020) and KITTI (Geiger et al., 2012), PointPlug improves every detector we tested under identical backbones and training settings.

The main contributions are summarized as follows:

- We formulate average-velocity transport for point sets by expressing the MeanFlow identity in a set-based, permutation-equivariant form, enabling teacher-free

one-step learning on 3D point clouds.

- We introduce 3DMF, which optimizes a two-time conditioned average-velocity field under a consistency objective and augments it with a lightweight shape-level constraint to preserve global shape, yielding stable one-step sampling.

- To our knowledge, we are the first to integrate one-step completion into modern 3D detection pipelines via PointPlug, a detector-agnostic module with negligible latency.

- 3DMF delivers high-fidelity completion and competitive generation, while offering substantial one-step sampling speedups (see Figure 1). With PointPlug, all evaluated detectors on nuScenes and KITTI improve under comparable settings.

## 2. Related Works

**Few-Step Flow Models.** Few-step generative modeling has been explored across different paradigms to accelerate sampling. Flow-based methods (Albergo & Vanden-Eijnden, 2023; Liu et al., 2023; Lipman et al., 2023; Frans et al., 2025) exploit continuous ODE dynamics and have produced a variety of acceleration schemes. Early approaches in diffusion (Luhman & Luhman, 2021; Salimans & Ho, 2022) distill many-step teachers into few-step students. Since direct compression of long, curved transport trajectories without controlling their geometry often harms one-step fidelity, rectified flows (Liu et al., 2023; Lipman et al., 2023; Albergo & Vanden-Eijnden, 2023) and reflow-based distillation (Wu et al., 2023b; Zhu et al., 2024; Kim et al., 2025) learn near-constant-velocity transport to straighten trajectories and mitigate this issue. In parallel, consistency models (Song et al., 2023; Song & Dhariwar, 2024; Geng et al., 2025b; Kim et al., 2024) supervise pairs of times to enable few-step jumps. MeanFlow (Geng et al., 2025a) establishes an average-velocity identity that enables teacher-free, from-scratch one-step training. Shortcut (Frans et al., 2025) further shows that step-size conditioning with self-consistency enables end-to-end few-step sampling. SCoT (Wu et al., 2025) unifies consistency and trajectory straightening under a single objective and is typically instantiated by distilling a pretrained diffusion model into a few-step generator. Inspired by MeanFlow (Geng et al., 2025a), we extend average-velocity transport to the 3D point clouds through 3DMF.

**Point Cloud Completion and Generation.** Completion and generation are both generative tasks: the former conditions on a partial scan, whereas the latter is unconditional or weakly conditioned (Lyu et al., 2022). Early completion methods (Yuan et al., 2018; Gadelha et al., 2018) were

driven by learned shape priors and typically adopted VAE-style encoder-decoder architectures. With modern generative modeling, diffusion-based (Luo & Hu, 2021; Zhou et al., 2021; Vahdat et al., 2022; Kasten et al., 2023; Ren et al., 2024) and flow-based (Yang et al., 2019; Kim et al., 2020) generators substantially improve completion fidelity but incur heavy sampling due to many denoising steps or ODE solves. Two routes dominate for reducing latency: distillation and rectified flows (Liu et al., 2023; Wu et al., 2023b; Zhu et al., 2024), but these approaches typically require multi-stage teacher pipelines. We instead train a teacher-free, one-step model from scratch for both conditional completion and unconditional generation, without pre-training, distillation, or curriculum learning.

**3D Object Detection.** 3D object detection is a widely studied downstream task on point clouds. Prior work has established strong LiDAR-only detectors (Liu et al., 2025; Zhou & Tuzel, 2018; Liu et al., 2024; Lang et al., 2019; Shi et al., 2020; Deng et al., 2021; Chen et al., 2023). To enhance performance, multimodal fusion is increasingly adopted: several methods (Wu et al., 2023c; Yu et al., 2025; Xie et al., 2023; Wu et al., 2023a; 2022) estimate depth from RGB and synthesize virtual points that are fused with LiDAR features to mitigate sparsity. LiDAR–radar fusion (Qian et al., 2021; Wang et al., 2023; Huang et al., 2025) exploits radar's resilience in rain, fog, and other adverse conditions to improve robustness. High-definition map priors provide static scene constraints (Casas et al., 2024; Zhang et al., 2024). Point cloud completion offers a single-sensor alternative to densify geometry, but prior work largely remains at the offline benchmark level (Yuan et al., 2018; Wu et al., 2023b; Cai et al., 2024). To our knowledge, this paper presents the first systematic evaluation of how point cloud completion affects 3D object detection.

## 3. 3D MeanFlow

We introduce 3D MeanFlow (3DMF), a one-step approach to point cloud completion and generation via average-velocity transport. Instead of fitting the instantaneous velocity, 3DMF learns an interval-averaged velocity field that collapses multi-step transport into a one-step update while preserving the global direction (see Figure 2). As point clouds lack canonical ordering, we enforce permutation equivariance by construction, making all mappings and objectives independent of point indexing.

**Goals.** Our goal is to achieve high-quality point cloud completion and generation in one step (with optional few-step refinement) using a simple, teacher-free training recipe, and to extend MeanFlow to unordered 3D by learning a permutation-equivariant average-velocity field.

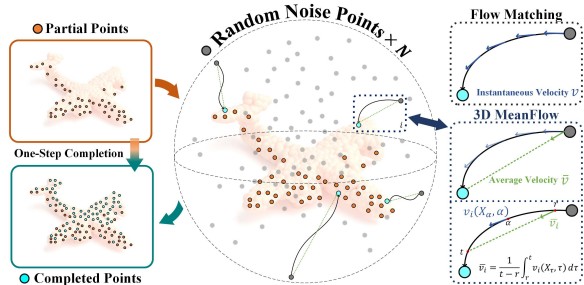

*Figure 2.* Illustration of 3D MeanFlow for one-step point cloud completion. Given partial points (orange), the model performs a one-step update (cyan) under the learned average-velocity field, transporting points toward the completed shape. Gray points are noise points used to visualize the transport field; we highlight one noise point $i$ at an intermediate time $\alpha \in [r, t]$ to contrast its instantaneous velocity $v_i(X_\alpha, \alpha)$ with its average-velocity vector $\bar{v}_i$ over $[r, t]$.

### 3.1. Preliminaries

**Flow Matching.** Let $x \sim p_{\text{data}}$ be a data sample and $\epsilon \sim p_{\text{prior}}$ be prior noise. Choose a time path from data to noise (e.g., linear mixture) to define intermediate state $z_t$ (Lipman et al., 2023; Liu et al., 2023; Albergo & Vanden-Eijnden, 2023). Since the same $z_t$ can arise from multiple $(x, \epsilon)$, Flow Matching learns the *marginal velocity*, the conditional expectation of the instantaneous velocity:

$$v(z_t, t) \triangleq \mathbb{E}_{p_t(x, \epsilon \mid z_t)}[v_t]. \tag{1}$$

At inference, we start from the prior and integrate $\dot{z}_t = v(z_t, t)$ forward in time. Intuitively, Flow Matching learns the instantaneous velocity field that specifies the local direction of motion.

**MeanFlow.** To compress long-interval transport into one or a few updates, we learn the *average velocity* $\bar{v}$ (Geng et al., 2025a), defined as total displacement over $[r, t]$ divided by its duration:

$$\bar{v}(z_t; r, t) \triangleq \frac{1}{t - r} \int_r^t v(z_\tau, \tau) \, \mathrm{d}\tau, \qquad r < t. \tag{2}$$

It satisfies the identity with the instantaneous velocity:

$$\bar{v}(z_t; r, t) = v(z_t, t) - (t - r) \frac{\mathrm{d}}{\mathrm{d}t} \bar{v}(z_t; r, t). \tag{3}$$

Conceptually, regressing $\bar{v}$ learns the integrated transport over $[r, t]$ rather than a local tangent field. By Equation (3), $\bar{v}$ equals the instantaneous field minus a finite-horizon correction scaled by $(t-r)$. This correction is absorbed into the training target, and as $t \to r$ the formulation reduces to standard Flow Matching. At inference, a single average-velocity transport step replaces numerical integration, avoiding cumulative discretization errors and enabling stable one-step completion and generation.

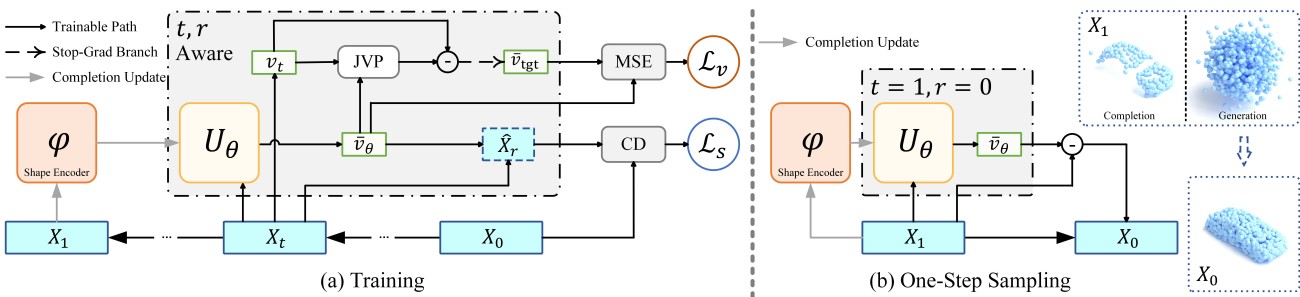

*Figure 3.* Overview of 3D MeanFlow. (a) Training optimizes two losses: a velocity MSE from a JVP-constructed stop-gradient target for $\bar{v}_\theta$, and a shape loss comparing the one-step output $\hat{X}_r$ to $X_0$. (b) One-step sampling maps $X_1$ (noise or partial input) to $X_0$ via a single update.

### 3.2. 3D MeanFlow Identity

We extend the MeanFlow identity to *unordered* point sets, making it applicable to 3D point clouds. Let $X_t \in \mathbb{R}^{N \times 3}$ denote the point cloud at time $t$. As point clouds lack canonical ordering, all mappings must be independent of point indexing. We therefore impose permutation equivariance on the velocity fields: for any permutation $\pi$ of point indices,

$$v(\pi X_t, t) = \pi \, v(X_t, t),$$
$$\bar{v}(\pi X_t; r, t) = \pi \, \bar{v}(X_t; r, t). \quad (4)$$

This aligns with standard set encoders and makes all objectives and mappings index-independent.

By Equation (2), the total displacement over $[r, t]$ equals $(t-r)\, \bar{v}(X_t; r, t)$. Hence, a single step of step length $(t-r)$ along $\bar{v}$ transports the state from $r$ to $t$.

Differentiating the average-velocity definition in Equation (2) with respect to $t$ along the trajectory $\dot{X}_t = v(X_t, t)$ yields the 3D MeanFlow identity for point clouds:

$$\bar{v}(X_t; r, t) \;=\; v(X_t, t) \;-\; (t-r)\frac{\mathrm{d}}{\mathrm{d}t}\bar{v}(X_t; r, t). \quad (5)$$

As $(t-r) \to 0$, the average velocity $\bar{v}(X_t; r, t)$ converges to the instantaneous velocity $v(X_t, t)$.

The total derivative of $\bar{v}$ along the trajectory decomposes into:

$$\frac{\mathrm{d}}{\mathrm{d}t}\bar{v}(X_t; r, t) = \underbrace{\partial_t \bar{v}(X_t; r, t)}_{\text{time term}}$$
$$+ \underbrace{\left(\nabla_X \bar{v}(X_t; r, t)\right) v(X_t, t)}_{\text{state term}}. \quad (6)$$

Here, $\nabla_X \bar{v}$ denotes the Jacobian with respect to all $3N$ coordinates of $X_t$ and $\dot{X}_t = v(X_t, t)$. The time term $\partial_t \bar{v}(X_t; r, t)$ measures the change in $\bar{v}$ when varying $t$ with $X_t$ held fixed, whereas the state term $\left(\nabla_X \bar{v}(X_t; r, t)\right) v(X_t, t)$ captures the change induced by the motion of $X_t$ along the trajectory $\dot{X}_t = v(X_t, t)$.

By Equation (4), both sides of Equation (6) transform equivariantly under any permutation, so the identity is well-defined for unordered point clouds.

**Conditional form.** In the conditional setting, the fields are conditioned on a fixed shape prior $s$. The identity becomes:

$$\bar{v}(X_t; r, t \mid s) = v(X_t, t \mid s) - (t-r)\frac{\mathrm{d}}{\mathrm{d}t}\bar{v}(X_t; r, t \mid s). \quad (7)$$

All permutation-equivariance statements carry over verbatim when conditioning on $s$. A detailed derivation is provided in Appendix A.

### 3.3. Training Objective

We train an average-velocity network $\mathcal{U}_\theta$ whose output $\bar{v}_\theta(X_t; r, t) \triangleq \mathcal{U}_\theta(X_t, r, t)$ is optimized to satisfy the 3D MeanFlow identity in expectation, and we add a shape-level consistency term to preserve global shape (see Figure 3(a)).

Throughout, $\| \cdot \|_2^2$ denotes the mean per-point squared $\ell_2$ norm over points and batch, which is permutation-invariant. For brevity, we suppress the arguments $(X_t; r, t)$ in $\bar{v}_\theta, \partial_t \bar{v}_\theta$, and $\nabla_X \bar{v}_\theta$ below.

**Instantaneous-Average consistency.** From Equation (6), the total derivative of $\bar{v}_\theta$ along the trajectory is:

$$D_t[\bar{v}_\theta] \;=\; \partial_t \bar{v}_\theta + \left(\nabla_X \bar{v}_\theta\right) v_t. \quad (8)$$

We define the velocity-consistency loss:

$$\mathcal{L}_v = \mathbb{E}_{(X_t, r, t)} \Big\| \bar{v}_\theta - \underbrace{\mathrm{sg}\big(v_t - (t-r)\, D_t[\bar{v}_\theta]\big)}_{\bar{v}_{\mathrm{tgt}}} \Big\|_2^2. \quad (9)$$

Here $\mathrm{sg}[\cdot]$ denotes the stop-gradient operator; its argument does not backpropagate into $\theta$. When $r = t$, the second term vanishes and the objective reduces to standard Flow Matching.

**Shape-Level consistency.** We define the one-step update as

$$\hat{X}_r \ = \ X_t - (t - r)\,\bar{v}_\theta(X_t; r, t), \qquad (10)$$

and compare it against a reference shape $X_r^{\mathrm{ref}}$ using the symmetric Chamfer distance:

$$\begin{aligned}
\mathcal{L}_s &= \mathrm{CD}\big(\hat{X}_r, \ X_r^{\mathrm{ref}}\big) \\
&= d\big(\hat{X}_r \,|\, X_r^{\mathrm{ref}}\big) \ + \ d\big(X_r^{\mathrm{ref}} \,|\, \hat{X}_r\big),
\end{aligned} \qquad (11)$$

with the directed term

$$d(A\,|\,B) \ = \ \frac{1}{|A|} \sum_{a \in A} \min_{b \in B} \|a - b\|_2. \qquad (12)$$

For generation, we set $X_r^{\mathrm{ref}} = X_0$; for completion, we set $X_r^{\mathrm{ref}}$ to the full ground-truth shape and compute $\hat{X}_r$ with a mask that updates only missing points.

This shape-level constraint penalizes overshoot in sparse or scale-varying regions, effectively regularizing the one-step displacement magnitude.

The full objective is

$$\mathcal{L} \ = \ \mathcal{L}_v \ + \ \lambda \mathcal{L}_s, \qquad (13)$$

where $\lambda$ is a hyperparameter.

### 3.4. One-Step / Few-Step Sampling

At inference, we apply a one-step operator induced by the average-velocity field. Given a state $X_t$ and a time pair $(r, t)$ with $r < t$, we define

$$\Phi_\theta\big(X_t; r, t, s\big) = X_t - (t - r)\,\bar{v}_\theta\big(X_t; r, t \,|\, s\big), \qquad (14)$$

where $s$ is a fixed shape prior (set $s = \varnothing$ for unconditional generation). For one-step sampling, set $(r, t) = (0, 1)$:

$$X_0 \ = \ \Phi_\theta\big(X_1; 0, 1, s\big), \qquad (15)$$

Few-step sampling partitions $[0, 1]$ into $K$ subintervals and composes $\Phi_\theta$ over the partition. See Figure 3(b) for a schematic.

For *unconditional generation*, initialize $X_1 \sim \mathcal{N}(0, I)$ and set $s = \varnothing$, yielding:

$$X_0 \ = \ \Phi_\theta\big(X_1; 0, 1, \varnothing\big) = X_1 - \bar{v}_\theta\big(X_1; 0, 1\big). \qquad (16)$$

For *conditional completion*, we obtain a shape prior from the observed points using $\varphi$ and denote it by $s$; we then initialize $X_1$ with Gaussian noise scaled to the dispersion of the observed points. This observation-scale normalization stabilizes initialization across scales. Then apply the conditional field:

$$X_0^{\mathrm{full}} \ = \ \Phi_\theta\big(X_1; 0, 1, s\big) = X_1 - \bar{v}_\theta\big(X_1; 0, 1 \,|\, s\big). \qquad (17)$$

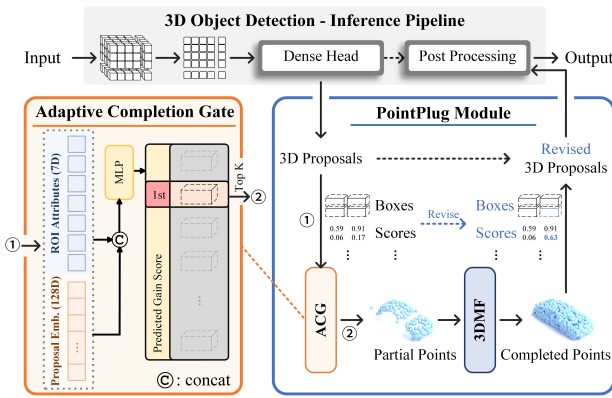

*Figure 4.* Overview of PointPlug integrated into a generic 3D detector. 3D proposals from the dense head go via ① to ACG, which scores each RoI using pre-completion features (7 RoI attributes and a 128-D proposal embedding) and selects the top-$K$ per frame. Selected RoIs then go via ② to 3DMF for one-step completion. The completed geometry is used to update boxes and class scores (without modifying the scene point cloud); the revised proposals then return to post-processing.

Thus generation and completion share the same operator $\Phi_\theta$; they differ only in the prior $s$ and in initialization, and both support one-step and few-step sampling within a unified framework.

## 4. PointPlug: 3DMF for 3D Object Detection

Prior completion methods are typically evaluated offline only. To quantify their impact on 3D detection, we introduce *PointPlug*, a minimal detector-agnostic module that integrates one-step completion into standard detectors (see Figure 4). PointPlug sits between proposal generation (dense or separate detection head) and post-processing: it scores proposals using pre-completion signals, completes up to $K$ proposals per frame in one step, and returns revised boxes and scores to the detector. Further analysis and implementation details are provided in Appendix B.

**Adaptive Completion Gate.** We construct a small offline Completion Impact Dataset (CID) that pairs pre-completion features with the observed one-step BEV-IoU gain per proposal, $g_i = \mathrm{IoU}_{\mathrm{BEV}}^{\mathrm{post}} - \mathrm{IoU}_{\mathrm{BEV}}^{\mathrm{pre}}$. A lightweight MLP takes 7 RoI attributes and a 128-D proposal embedding and outputs a predicted gain score. At inference, given a per-frame completion cap $K$, we rank proposals, complete the top-$K$, and update their class scores with a non-decreasing fusion of the detector and ACG scores. The remaining proposals retain the detector scores. From CID we find that positive gains are sparse at the frame level: an oracle top-1 recovers about 38% of the positive headroom and top-2 about $50\%$. Marginal returns drop sharply with $K$, by $K = 3$ the average gain is negligible.

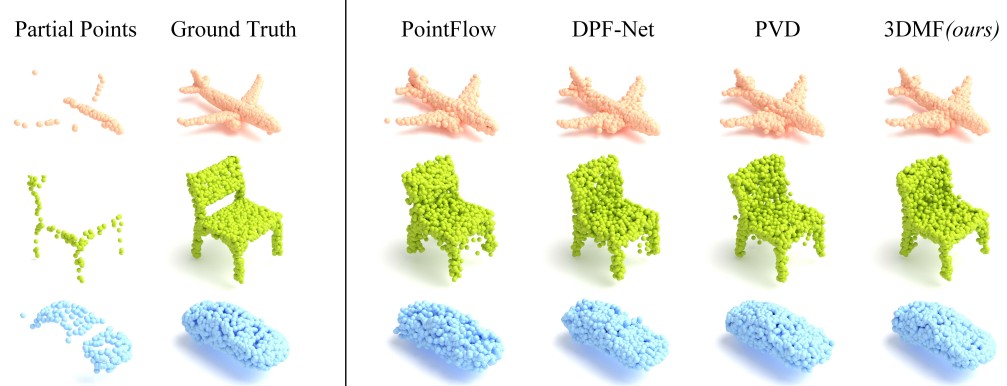

Partial Points    Ground Truth      PointFlow      DPF-Net      PVD      3DMF *(ours)*

*Figure 5.* Point cloud completion comparison, all methods output 2048 points. Results for 3DMF use the one-step setting (3DMF-1NFE). Our method produces more faithful shapes with cleaner surfaces than representative baselines.

*Table 1.* Completion results on Airplane, Chair, Car compared with baselines. We report CD↓ ($\times 10^3$), EMD↓ ($\times 10^2$), and sampling time per shape (s). 3DMF-5NFE attains the best CD on all three categories, and 3DMF-1NFE achieves the fastest sampling time.

| Model | Airplane | | Chair | | Car | | Sampling Time (s) |
|---|---|---|---|---|---|---|---|
| | CD↓ | EMD↓ | CD↓ | EMD↓ | CD↓ | EMD↓ | |
| PointFlow (Yang et al., 2019) | 0.403 | 1.180 | 2.707 | 3.649 | 1.803 | 2.851 | 0.27 |
| SoftFlow (Kim et al., 2020) | 0.404 | 1.198 | 2.786 | 3.295 | 1.850 | 2.789 | 0.12 |
| PVD (Zhou et al., 2021) | 0.442 | 1.030 | 3.211 | 2.939 | 1.774 | 2.146 | 29.9 |
| DPF-Net (Shuai et al., 2023) | 0.528 | 1.105 | 2.763 | 3.320 | 1.396 | 2.318 | 0.34 |
| PointNSP (Meng et al., 2025) | 0.401 | **1.008** | 2.702 | **2.878** | 1.384 | 2.068 | 5.48 |
| **3DMF-1NFE** *(ours)* | 0.639 | 1.167 | 2.933 | 3.541 | 1.224 | 1.896 | **0.005** |
| **3DMF-5NFE** *(ours)* | **0.369** | 1.121 | **2.138** | 3.188 | **0.975** | **1.752** | 0.013 |

**PointPlug Pipeline.** For a selected proposal with box $b = (x, y, z, \ell, w, h, \psi)$, we crop points inside $b$, translate to its center, and apply isotropic normalization with $s = \max\{\ell, w, h\}$. Set a target count $N$; subsample if the visible points exceed $N$, otherwise repeat visible points to reach $N$. Feed the fixed-size partial set to a one-step 3DMF update in the canonical frame; then de-normalize to world coordinates and combine with the observed RoI points only to refit the box. Finally, fuse ACG and detector scores for the completed RoIs and pass the updated proposals to post-processing. The backbone and heads remain unchanged; inference remains synchronous.

## 5. Experiments

### 5.1. Point Cloud Completion

3DMF delivers strong completion quality with millisecond-level latency, with the best accuracy in the few-step setting. We report two inference settings, 1NFE (one step) and 5NFE (five steps), where NFE denotes the number of 3DMF update steps per sample.

**Settings.** We follow the setup in prior work (Zhang et al., 2018; Zhou et al., 2021; Wu et al., 2023c). For each ShapeNet (Chang et al., 2015) shape, we render 20 ran-

dom views, back-project depths, and uniformly sample 200 visible points as the partial input. For every ground-truth shape, completion is evaluated on all 20 partials and report the mean over these trials. We apply the same normalization and post-processing across our evaluations; outputs contain 2,048 points. We report Chamfer Distance (CD↓) and Earth Mover's Distance (EMD↓). Results are given on the three largest benchmark categories: Airplane, Chair, and Car. We train a separate model for each category, and report results on the corresponding test split.

For 3DMF, sampling times are per-shape forward passes with batch size 1 on an NVIDIA A800. Baseline sampling times are cross-referenced from the corresponding original papers, following the convention of prior work (Wu et al., 2023c). Under matched hardware, 3DMF-1NFE remains the fastest by a large margin (see Appendix C.3). For further details on the implementation, refer to Appendix C.

**Quantitative Results.** Table 1 summarizes the results. 3DMF-5NFE attains the best CD on Airplane, Chair, and Car, while maintaining millisecond-level latency. For EMD, results vary by category: thin or highly articulated structures tend to yield higher EMD, while others are lower. This reflects EMD's sensitivity to fine-grained mass transport, whereas our step-limited objective prioritizes global

shape consistency captured by CD. 3DMF-1NFE remains millisecond-level and offers an attractive speed-quality trade-off.

**Qualitative Results.** We compare completions under identical partial inputs (see Figure 5). At 1NFE and 5NFE, 3DMF recovers the global structure and produces sharper boundaries and finer details than representative baselines.

These results show that 3DMF sustains high sampling efficiency and supports real-time point cloud completion without sacrificing fidelity.

### 5.2. Point Cloud Generation

We evaluate 3DMF on unconditional generation to test the one-step paradigm without conditioning. In this setting, 3DMF attains competitive quality while retaining an order-of-magnitude speed advantage.

**Settings.** We follow the completion setup (Section 5.1) and previous work (Zhang et al., 2018; Zhou et al., 2021; Wu et al., 2023c) on ShapeNet (Chang et al., 2015). We generate shapes with 2,048 points for Airplane, Chair, and Car categories. For evaluation, we use the robust one-nearest-neighbor accuracy (1-NNA) computed with CD and EMD. See Appendix C for further experimental settings and additional generation metrics.

**Results.** Table 2 reports generation quality and sampling time. 3DMF-1NFE remains the fastest among all methods. In the unconditional setting, few-step models may reduce sample diversity, so gains are less pronounced than in completion. However, 3DMF-5NFE achieves the best results in the Car category. Qualitative comparisons at 1NFE are shown in Figure 10. Additional MMD (Matching Distance) and COV (Coverage Score) results are provided in Appendix C and Table 12.

### 5.3. 3D Object Detection with One-Step Completion

To quantify how completion benefits detectors, we insert PointPlug into inference pipelines and evaluate end-to-end performance on real LiDAR point clouds from nuScenes (Caesar et al., 2020) and KITTI (Geiger et al., 2012). This setting also tests completion in real-world driving scenes. Additional results and implementation details are provided in Appendix D.

**Settings.** We evaluate on nuScenes and KITTI. Baselines are FSHNet (Liu et al., 2025) and VoxelNeXt (Chen et al., 2023) on nuScenes, and PV-RCNN (Shi et al., 2020) and Voxel-RCNN (Deng et al., 2021) on KITTI. PointPlug is placed immediately before post-processing—after the dense head for FSHNet / VoxelNeXt and after the RoI head for

*Table 2.* Category-wise point cloud generation comparisons. We report 1-NNA (%) computed with CD and EMD. Values closer to 50% indicate better fidelity and diversity. 3DMF (1NFE / 5NFE) delivers competitive quality at the fastest sampling rate.

| Category | Model | 1-NNA (%) | | Time (s) |
|---|---|---|---|---|
| | | CD↓ | EMD↓ | |
| Airplane | PointFlow (2019) | 75.68 | 70.74 | 0.27 |
| | DPF-Net (2023) | 75.18 | 65.55 | 0.33 |
| | PVD (2021) | 73.82 | 64.81 | 29.9 |
| | PSF (2023c) | **71.11** | **61.09** | 0.04 |
| | PointNSP (2025) | 72.24 | 63.69 | 5.48 |
| | **3DMF-1NFE** *(ours)* | 76.19 | 70.27 | **0.005** |
| | **3DMF-5NFE** *(ours)* | 73.38 | 64.32 | 0.013 |
| Chair | PointFlow (2019) | 62.84 | 60.57 | 0.27 |
| | DPF-Net (2023) | 62.00 | 58.53 | 0.33 |
| | PVD (2021) | 56.26 | 53.32 | 29.9 |
| | PSF (2023c) | 58.92 | 54.45 | 0.04 |
| | PointNSP (2025) | **54.54** | **52.85** | 5.48 |
| | **3DMF-1NFE** *(ours)* | 58.73 | 55.01 | **0.005** |
| | **3DMF-5NFE** *(ours)* | 56.55 | 53.70 | 0.013 |
| Car | PointFlow (2019) | 58.10 | 56.25 | 0.27 |
| | DPF-Net (2023) | 62.35 | 54.48 | 0.33 |
| | PVD (2021) | 54.55 | 53.83 | 29.9 |
| | PSF (2023c) | 57.19 | 56.07 | 0.04 |
| | PointNSP (2025) | 52.17 | 51.85 | 5.48 |
| | **3DMF-1NFE** *(ours)* | 54.03 | 53.24 | **0.005** |
| | **3DMF-5NFE** *(ours)* | **52.06** | **51.81** | 0.013 |

*Table 3.* Detection performance on the nuScenes *val* set. NDS, mAP, and Car mAP by range (0-30m, 30-50m, 50m+). Suffix *-pp* denotes the detector with the PointPlug module inserted. One-step completion yields consistent improvements overall, with clear relative gains at 30-50m and the strongest gains at 50m+.

| Method | NDS | mAP | Car mAP | | | Time |
|---|---|---|---|---|---|---|
| | | | 0–30m | 30–50m | 50m+ | (ms) |
| VoxelNeXt (2023) | 68.7 | 63.5 | 91.87 | 68.89 | 12.35 | 63 |
| **VoxelNeXt-pp** | 68.9 | 64.0 | 91.90 | 71.74 | 18.31 | 69 |
| FSHNet (2025) | 71.7 | 68.1 | 95.81 | 72.82 | 16.23 | 123 |
| **FSHNet-pp** | 71.8 | 68.5 | 95.83 | 75.43 | 23.85 | 129 |

PV-RCNN / Voxel-RCNN.

We use 3DMF-1NFE trained on the *Car* category and cap completion at $K=2$ RoIs per frame. Since only cars are completed, we report both the official aggregates and Car-only metrics. We report end-to-end per-frame latency; completion is run once per frame in batch over the selected RoIs.

For nuScenes, we additionally stratify results by range (0–30m, 30–50m, 50m+). Note that official nuScenes metrics consider objects within 50m, so improvements beyond 50m are not reflected in the aggregates.

**Proposal dependency.** PointPlug operates on detector proposals by completing points within selected RoIs before

*Table 4.* Detection performance on the KITTI *test* set. Suffix *-pp* denotes the detector with the PointPlug module inserted.

| Method | Car 3D AP (R40) | | | Car BEV AP (R40) | | |
|---|---|---|---|---|---|---|
| | Easy | Mod. | Hard | Easy | Mod. | Hard |
| PV-RCNN (2020) | 90.25 | 81.43 | 76.82 | 94.98 | 90.65 | 86.14 |
| **PV-RCNN-pp** | 90.36 | 82.12 | 78.91 | 95.01 | 91.20 | 87.53 |
| Voxel-RCNN (2021) | 90.90 | 81.62 | 77.06 | 94.85 | 88.83 | 86.13 |
| **Voxel-RCNN-pp** | 91.03 | 82.40 | 79.24 | 94.91 | 89.58 | 87.60 |

*Table 5.* Time sampler ablation on *Car* completion. $t$ and $r$ are sampled from the listed distributions. $^{\dagger}$ denotes a high-$t$ mixture that oversamples the noise end.

| $t$ sampler | $r$ sampler | 1NFE | | 5NFE | |
|---|---|---|---|---|---|
| | | CD↓ | EMD↓ | CD↓ | EMD↓ |
| uniform(0, 1) | uniform(0, 1) | 3.525 | 3.112 | 2.354 | 2.159 |
| lognorm(-0.4, 1) | lognorm(-0.4, 1) | 2.471 | 2.841 | 1.781 | 2.231 |
| uniform(0, 1)$^{\dagger}$ | uniform(0.65t, t) | 1.529 | 2.442 | 1.983 | 3.679 |
| beta(2, 2)$^{\dagger}$ | uniform(0.65t, t) | 1.224 | 1.896 | 0.975 | 1.752 |
| beta(2, 2)$^{\dagger}$ | uniform(0.5t, t) | 1.318 | 1.861 | 1.023 | 1.640 |
| beta(2, 5)$^{\dagger}$ | uniform(0.65t, t) | 1.473 | 3.023 | 0.972 | 2.043 |

*Table 6.* Shape consistency weight ablation on *Car* completion. $\lambda$ is the weight of the shape consistency loss. We report CD and EMD under 1NFE and 5NFE.

| $\lambda$ | 1NFE | | 5NFE | |
|---|---|---|---|---|
| | CD↓ | EMD↓ | CD↓ | EMD↓ |
| 0.0 | 1.529 | 2.401 | 1.293 | 1.986 |
| 0.1 | 1.395 | 2.229 | 1.072 | 1.880 |
| 1.0 | 1.283 | 1.917 | 1.051 | 1.873 |
| 2.0 | 1.224 | 1.896 | 0.975 | 1.752 |
| 5.0 | 1.431 | 2.698 | 1.184 | 2.040 |

*Table 7.* Effect of the per-frame completion cap $K$ in PointPlug on the nuScenes *val* set. $K$ is the maximum number of RoIs completed per frame. We report detection performance bucketed by range and per-frame inference time.

| $K$ | NDS | mAP | Car mAP | | | Time |
|---|---|---|---|---|---|---|
| | | | 0–30m | 30–50m | 50m+ | (ms) |
| 0 | 71.7 | 68.1 | 95.81 | 72.82 | 16.23 | 123 |
| 1 | 71.7 | 68.3 | 95.81 | 74.31 | 20.20 | 127 |
| 2 | 71.8 | 68.5 | 95.83 | 75.43 | 23.85 | 129 |
| 3 | 71.6 | 67.9 | 95.76 | 71.85 | 19.89 | 130 |

post-processing; it therefore cannot address false negatives with no proposals. We discuss this limitation and possible extensions in Appendix D.1.

**Results on nuScenes.** As shown in Table 3, one-step completion produces clear gains for long-range cars. Car mAP increases markedly in the 50m+ bin ($\approx 45 - 50\%$ relative), with smaller but consistent gains at 30-50m, while near range remains stable. The official NDS/mAP averages only over the 0-50m scope and across all classes; thus aggregate changes are modest despite the long-range gains.

**Results on KITTI.** On KITTI (Geiger et al., 2012) (Table 4), we observe a similar pattern: gains are most pronounced on the *Moderate* and *Hard* splits, where occlusion and sparsity are severe. Improvements on *Easy* are smaller but consistent.

These results indicate that one-step completion is effective on real LiDAR point clouds and provides targeted gains for sparse or distant objects.

### 5.4. Ablation Study

**Ablations for 3DMF.** We analyze key 3DMF hyperparameters using Car completion as the main case. We focus on two departures from the original MeanFlow: the time sampler and the weight of the shape-level consistency term. A complete set of ablations is provided in Appendix E.

Table 5 compares time samplers for $(t, r)$. Following Mean-Flow (Geng et al., 2025a), we test a uniform and a logit-normal sampler with $t > r$ enforced by swapping. Both

underperform, especially at 1NFE. We first draw $t$, then sample $r$ from a linear interval bounded above by $t$. This design ties the update horizon to the noise level. At high-noise times ($t \approx 1$), set $r$ close to t to limit transport; at low-noise times ($t \approx 0$), widen the interval to allow stronger corrections. Rows marked with $^{\dagger}$ use a high-$t$ mixture: with a fixed probability they oversample the noise end; otherwise they draw from a broad base distribution, improving adaptation to high-noise states.

To promote global shape coherence, we add a shape-level consistency term to the loss. We vary its weight $\lambda$ in Table 6. The best trade-off occurs at $\lambda = 2.0$. With $\lambda = 0$, the model reduces to the MeanFlow baseline. If $\lambda$ is too large, the average-velocity transport objective is diluted and the model collapses toward the centroid, degrading completion quality.

**Per-Frame Completion Cap for 3D Detection.** We vary the per-frame cap $K$ defined in Section 4 and evaluate FSHNet-pp on the nuScenes val set. As shown in Table 7, increasing $K$ from 0 to 1-2 improves detection, with the largest gains at 30-50m and 50m+. Beyond $K=2$, marginal returns saturate and near-range performance can degrade. We attribute this decline to mis-completions induced by a larger cap, which admits lower-confidence RoIs. Since $K$ is an upper bound, many frames complete fewer than $K$ RoIs, so added time does not grow linearly with $K$.

## 6. Conclusion

In this paper, we present 3DMF, a teacher-free, one-step approach to point cloud completion and generation via average-

velocity transport that maintains high fidelity. Additionally, we introduce PointPlug, a detector-agnostic plug-in module that integrates one-step completion into 3D detection pipelines and quantifies its impact. Extensive experiments show strong completion performance, competitive generation, and consistent improvements across evaluated detectors. Future work includes improving diversity in one-step generation and further optimizing deployment for real-world perception systems.

## Acknowledgements

This work was supported in part by National Natural Science Foundation of China under Grant No. 62272344, in part by the Natural Science Foundation of Anhui under Grant No. 2508085QF241. The authors are grateful to the Area Chairs and the anonymous reviewers for their constructive comments.

## Impact Statement

This work advances one-step 3D point cloud completion and generation, and introduces PointPlug for completion-driven refinement in 3D detection. These advances may support efficient and robust 3D understanding and content creation across domains such as robotics, autonomous driving, and graphics.

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

## Appendix Overview

In the Appendix, we provide additional details organized as follows:

1. Appendix A: Additional Derivations for 3D MeanFlow.

2. Appendix B: Additional Technical Details of PointPlug.

3. Appendix C: Completion and Generation: Training Details and Additional Results.

4. Appendix D: PointPlug: Proposal Dependency Analysis and Extended Detection Results.

5. Appendix E: Additional Ablations and Analyses.

6. Appendix F: Additional Qualitative Results.

## A. Additional Derivations for 3D MeanFlow

### A.1. Permutation Equivariance for Point Sets

Point clouds are unordered sets, so any relabeling of point indices must not change the semantics of the model. To make the 3D formulation well defined on sets, we require permutation *equivariance* for vector-valued mappings such as velocity fields, and permutation *invariance* for set summaries such as global descriptors and losses. Let $S_N$ be the permutation group on $N$ points and $P_\pi \in \{0,1\}^{N \times N}$ be the permutation matrix of $\pi \in S_N$. The action on a point cloud $X \in \mathbb{R}^{N \times 3}$ is

$$\pi X \triangleq P_\pi X. \tag{18}$$

A mapping $F : \mathbb{R}^{N \times d} \to \mathbb{R}^{N \times d'}$ is *permutation-equivariant* if

$$F(\pi X) = \pi F(X) \qquad \forall \pi \in S_N, \tag{19}$$

while $G : \mathbb{R}^{N \times d} \to \mathbb{R}^{d'}$ is *permutation invariant* if

$$G(\pi X) = G(X) \qquad \forall \pi \in S_N. \tag{20}$$

Equivariance means that relabeling the inputs only relabels the outputs in the same way; invariance means global summaries are unaffected by relabeling.

**Equivariance of Velocity Fields and Derivatives.** As in the main text, let $X_t \in \mathbb{R}^{N \times 3}$ denote the point cloud state at time $t$, and let $v(X_t, t)$ and $\bar{v}(X_t; r, t)$ denote the instantaneous and interval-averaged velocity fields. We impose

$$v(\pi X_t, t) = \pi v(X_t, t), \tag{21}$$

$$\bar{v}(\pi X_t; r, t) = \pi \bar{v}(X_t; r, t). \tag{22}$$

Write $\nabla_X \bar{v} \in \mathbb{R}^{(3N) \times (3N)}$ for the Jacobian with respect to $X$ (stacked over points), and $\partial_t \bar{v}$ for the partial derivative with $X$ fixed. Linearity of $P_\pi$ and the chain rule yield

$$\partial_t \bar{v}(\pi X_t; r, t) = \pi \partial_t \bar{v}(X_t; r, t), \tag{23}$$

$$\nabla_X \bar{v}(\pi X_t; r, t) = P_\pi \nabla_X \bar{v}(X_t; r, t) P_\pi^\top. \tag{24}$$

Combining Equations (21) to (24) gives the permutation-covariant form of the state term in the total derivative:

$$\left( \nabla_X \bar{v}(\pi X_t; r, t) \right) v(\pi X_t, t) = P_\pi \left( \nabla_X \bar{v}(X_t; r, t) \right) v(X_t, t). \tag{25}$$

Therefore, the total derivative transforms equivariantly under permutations:

$$\frac{\mathrm{d}}{\mathrm{d}t} \bar{v}(\pi X_t; r, t) = \pi \frac{\mathrm{d}}{\mathrm{d}t} \bar{v}(X_t; r, t), \tag{26}$$

which ensures that the 3D MeanFlow identity in the main text is well defined for unordered point sets.

**Equivariance of the One-Step Operator.** Consider the one-step 3DMF operator from the main text,

$$\Phi_\theta(X_t; r, t, s) = X_t - (t - r)\,\bar{v}_\theta(X_t; r, t \mid s), \tag{27}$$

permutation equivariance of $\bar{v}_\theta$ immediately implies

$$\begin{aligned}
\Phi_\theta(\pi X_t; r, t, s) &= \pi X_t - (t - r)\,\bar{v}_\theta(\pi X_t; r, t \mid s) \\
&= \pi X_t - (t - r)\,\pi\,\bar{v}_\theta(X_t; r, t \mid s) \\
&= \pi\,\Phi_\theta(X_t; r, t, s).
\end{aligned} \tag{28}$$

Thus $\Phi_\theta$ is permutation-equivariant, and the one-step 3DMF update is well defined on unordered point sets.

**Conditioning via a Permutation-Invariant Prior.** In the conditional setting, a shape prior $s$ is computed from the observed subset $X_p$ by a permutation-invariant encoder $\varphi$:

$$\varphi(\pi X_p) = \varphi(X_p), \qquad s = \varphi(X_p). \tag{29}$$

Treating $s$ as fixed, Equations (21) and (22)—and hence Equations (23) and (26)—remain valid; conditioning does not break the set structure.

**Compatibility of Training Objectives with Permutations.** The Chamfer distance used in the shape-level loss is permutation-invariant since nearest-neighbor queries and pointwise averaging do not depend on point labels:

$$\begin{aligned}
\mathrm{CD}(P_\pi A, B) &= \mathrm{CD}(A, B), \\
\mathrm{CD}(A, P_\pi B) &= \mathrm{CD}(A, B).
\end{aligned} \tag{30}$$

The velocity consistency loss is the mean of per-point squared errors,

$$\mathcal{L}_v = \frac{1}{N}\sum_{i=1}^{N}\left\|\bar{v}_{\theta,i} - \bar{v}_{\mathrm{tgt},i}\right\|_2^2, \tag{31}$$

and is invariant under any common permutation of predictions and targets.

For completion, the constraint of keeping observed points fixed can be expressed with a mask $M \in \{0,1\}^N$ (1 means observed) and the Hadamard product $\odot$:

$$X_r = X_t - (t - r)\left[(\mathbf{1} - M) \odot \bar{v}_\theta(X_t; r, t \mid s)\right]. \tag{32}$$

If $(X_t, M)$ are permuted simultaneously, then

$$X_r(\pi X_t, P_\pi M) = P_\pi X_r(X_t, M), \tag{33}$$

so the masking update is consistent with permutation equivariance.

Finally, Jacobian–vector products and total-derivative terms in the velocity objective inherit the same covariance. If $w(\cdot)$ is a permutation-equivariant vector field,

$$\left(\nabla_X \bar{v}_\theta(\pi X)\right) w(\pi X) = P_\pi\left(\nabla_X \bar{v}_\theta(X)\, w(X)\right), \tag{34}$$

which ensures that all differential terms in the training computation graph respect the permutation structure of the point set.

**Summary.** With these properties, the 3DMF identity and the associated training and sampling operators are well defined on unordered point sets: velocity fields and their derivatives transform equivariantly, the one-step operator is equivariant, conditioning via a permutation-invariant prior preserves the permutation structure, and the loss terms are compatible with permutations.

## A.2. Derivation of the 3D MeanFlow Identity

We use the same notation and assumptions as in the main text. The point cloud state is $X_t \in \mathbb{R}^{N \times 3}$ at time $t$, the instantaneous velocity field is $v(X_t, t)$, and the interval-averaged field is $\bar{v}(X_t; r, t)$. Permutation equivariance holds throughout, and unless stated otherwise, $r$ is treated as fixed.

**Definition and Displacement.** The average velocity over $[r, t]$ is defined as:

$$\bar{v}(X_t; r, t) \triangleq \frac{1}{t - r} \int_r^t v(X_\tau, \tau) \, \mathrm{d}\tau, \qquad r < t. \tag{35}$$

Introduce the displacement over $[r, t]$ as:

$$S(X_t; r, t) \triangleq (t - r) \, \bar{v}(X_t; r, t), \tag{36}$$

which, by Equation (35), equals the time integral of the instantaneous field,

$$S(X_t; r, t) = \int_r^t v(X_\tau, \tau) \, \mathrm{d}\tau. \tag{37}$$

**Differentiation and the 3D MeanFlow Identity.** Differentiating Equation (37) with respect to $t$ gives $v(X_t, t)$ by the Fundamental Theorem of Calculus. Differentiating Equation (36) and using the product rule yields:

$$\bar{v}(X_t; r, t) + (t - r) \frac{\mathrm{d}}{\mathrm{d}t} \bar{v}(X_t; r, t) = v(X_t, t). \tag{38}$$

Rearranging Equation (38) gives the 3D MeanFlow identity,

$$\bar{v}(X_t; r, t) = v(X_t, t) - (t - r) \frac{\mathrm{d}}{\mathrm{d}t} \bar{v}(X_t; r, t). \tag{39}$$

**Total Derivative.** Define the total derivative operator along the trajectory $\dot{X}_t = v(X_t, t)$ as

$$\begin{aligned} D_t[\bar{v}](X_t; r, t) &\triangleq \partial_t \bar{v}(X_t; r, t) \\ &\quad + \left( \nabla_X \bar{v}(X_t; r, t) \right) v(X_t, t). \end{aligned} \tag{40}$$

Then $\frac{\mathrm{d}}{\mathrm{d}t} \bar{v}(X_t; r, t) = D_t[\bar{v}](X_t; r, t)$, and Equation (39) becomes the compact form

$$\bar{v}(X_t; r, t) = v(X_t, t) - (t - r) D_t[\bar{v}](X_t; r, t). \tag{41}$$

For completeness, the expanded form used to construct the velocity target is:

$$\bar{v}(X_t; r, t) = v(X_t, t) - (t - r) \partial_t \bar{v}(X_t; r, t) - (t - r) \left( \nabla_X \bar{v}(X_t; r, t) \right) v(X_t, t). \tag{42}$$

Regressing $\bar{v}$ amounts to learning the integrated transport over $[r, t]$; the finite-horizon correction $(t - r) D_t[\bar{v}](X_t; r, t)$ scales linearly with $(t - r)$ and can be computed with a single Jacobian–vector product (JVP) when forming the stop-gradient target.

**Small-Interval Limit and Additivity.** As $(t - r) \to 0$, Equation (39) implies

$$\lim_{t \to r} \bar{v}(X_t; r, t) = v(X_t, t). \tag{43}$$

Moreover, by additivity of the integral in Equation (37), for any $s \in (r, t)$ we have

$$(t - r) \, \bar{v}(X_t; r, t) = (s - r) \, \bar{v}(X_s; r, s) + (t - s) \, \bar{v}(X_t; s, t). \tag{44}$$

Equation (44) shows that a long-step displacement decomposes into two shorter steps, a consistency property inherited directly from the definition.

**One-Step Update.** Finally, Equations (36) and (37) imply the one-step update used at sampling time

$$X_r = X_t - (t - r)\,\bar{v}(X_t; r, t), \tag{45}$$

and partitioning $[0, 1]$ into $K$ subintervals yields few-step sampling by iterating Equation (45).

## A.3. Conditional 3D MeanFlow Identity with Shape Prior

Our formulation also supports conditioning. The main text states the conditional identity; here we provide its derivation details and a permutation-invariant construction of the shape prior.

**Permutation-Invariant Shape Prior.** Given the observed subset $X_p$, we compute a shape prior using a permutation-invariant encoder $\varphi$:

$$\varphi(P_\pi X_p) = \varphi(X_p), \qquad s = \varphi(X_p). \tag{46}$$

A concrete instantiation uses a symmetric pooling operator $\rho$ over pointwise encodings, together with learnable maps $\phi$ and $\psi$:

$$s = \psi\big(\rho(\{\phi(x)\}_{x \in X_p})\big). \tag{47}$$

**Conditional 3D MeanFlow Identity.** With the prior $s$ computed as in Equations (46) and (47) and treated as a fixed input, we write the conditional instantaneous and interval-averaged fields as $v(X_t, t \,|\, s)$ and $\bar{v}(X_t; r, t \,|\, s)$. The conditional 3D MeanFlow identity is

$$\bar{v}(X_t; r, t \,|\, s) = v(X_t, t \,|\, s) - (t - r)\frac{\mathrm{d}}{\mathrm{d}t}\,\bar{v}(X_t; r, t \,|\, s), \tag{48}$$

with total derivative

$$\frac{\mathrm{d}}{\mathrm{d}t}\,\bar{v}(X_t; r, t \,|\, s) = \partial_t \bar{v}(X_t; r, t \,|\, s) + \big(\nabla_X \bar{v}(X_t; r, t \,|\, s)\big)\,v(X_t, t \,|\, s). \tag{49}$$

Equivalently, using $D_t[\cdot]$ from Equation (40),

$$\bar{v}(X_t; r, t \,|\, s) = v(X_t, t \,|\, s) - (t - r)\,D_t[\bar{v}](X_t; r, t \,|\, s). \tag{50}$$

**Conditional Training Target.** Let the network output be $\bar{v}_\theta(X_t; r, t \,|\, s)$. The stop-gradient target is

$$\bar{v}_{\mathrm{tgt}}(X_t; r, t \,|\, s) = v(X_t, t \,|\, s) - (t - r)\,D_t[\bar{v}_\theta](X_t; r, t \,|\, s), \tag{51}$$

where

$$D_t[\bar{v}_\theta](X_t; r, t \,|\, s) \triangleq \partial_t \bar{v}_\theta(X_t; r, t \,|\, s) + \big(\nabla_X \bar{v}_\theta(X_t; r, t \,|\, s)\big)\,v(X_t, t \,|\, s). \tag{52}$$

The per-point MSE loss is permutation-invariant:

$$\mathcal{L}_v = \mathbb{E}_{(X_t, r, t, s)}\Big[\big\|\bar{v}_\theta(X_t; r, t \,|\, s) - \mathrm{sg}\big(\bar{v}_{\mathrm{tgt}}(X_t; r, t \,|\, s)\big)\big\|_2^2\Big]. \tag{53}$$

**Sampling and Mask-Constrained Update.** One-step sampling uses

$$X_r = X_t - (t - r)\,\bar{v}_\theta(X_t; r, t \,|\, s). \tag{54}$$

For completion with a binary mask $M \in \{0, 1\}^N$ that marks observed points, we enforce a mask-constrained update:

$$X_r = X_t - (t - r)\Big[(\mathbf{1} - M) \odot \bar{v}_\theta(X_t; r, t \,|\, s)\Big]. \tag{55}$$

**Equivariance Consistency.** If $\varphi$ is permutation-invariant as in Equation (46) and the fields satisfy the permutation properties in Equations (21) to (23) and (26), then Equations (48) to (52), (54) and (55) are compatible with permutations on unordered point sets.

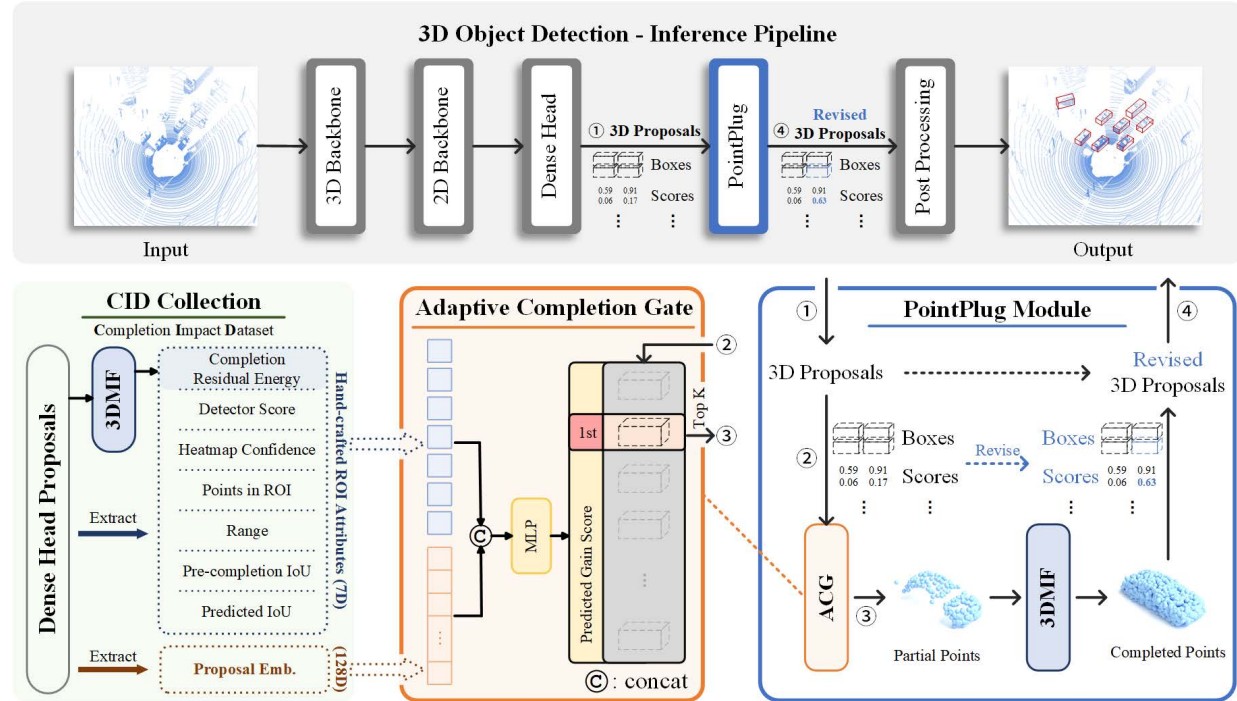

*Figure 6.* Overview of PointPlug integrated into a generic 3D detector. Dense head proposals go via ① to PointPlug, which extracts 7 RoI attributes and a 128-dimensional embedding; the features go via ② to ACG, which ranks proposals and selects the top-$K$ RoIs. For each selected RoI, its partial points go via ③ to 3DMF for one-step completion. The completed geometry revises boxes and class scores, then returns via ④ to post processing. Bottom-left: Completion Impact Dataset (CID) used to train ACG, which pairs these features with one-step completion gains.

## B. Additional Technical Details of PointPlug

This section provides additional implementation and technical details for PointPlug, and a complete schematic is provided in Figure 6. We begin with a brief completion impact analysis to determine where completion should be inserted in the detection pipeline and how this insertion affects performance at both the frame and proposal levels. We then describe how the proposal-level Completion Impact Dataset (CID) is constructed, which pre-completion signals it records, and the exact formulas for the seven RoI attributes, including their standardization and the concatenated input used by the Adaptive Completion Gate (ACG). Finally, we outline the canonicalization, one-step 3DMF update, and box refitting procedure used to compute the per-proposal BEV-IoU change. Throughout this section, $K$ denotes the per-frame completion budget.

### B.1. Completion Impact Analysis

To integrate one-step point cloud completion into 3D detection, we analyze where to insert completion in the pipeline, and what benefits and risks this insertion introduces.

**Where to Plug Completion.**    We evaluate two integration strategies:

(i) *Scene-level insertion:* Fuse the completed points back into the full scene and run the detector a second time. In practice, this breaks real-time operation and, under over-completion (e.g., hallucinating vehicle-like geometry where none exists), can propagate errors into other classes during the second pass.

(ii) *Proposal-level insertion:* Run completion only on dense-head proposals, then refine those proposals and continue the original detection pipeline. We adopt the proposal-level design to bound latency and to localize any potential side effects.

**Per-Proposal Completion Impact.**    We construct a per-proposal *Completion Impact Dataset* (CID) on the *nuScenes* (Caesar et al., 2020) using a strong baseline detector, FSHNet (Liu et al., 2025). For each frame, we enumerate dense-head

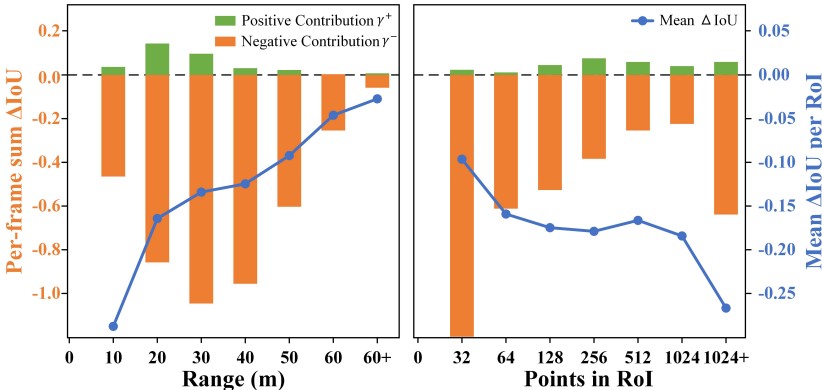

*Figure 7.* Completion impact across range and point count. Upward bars show the per-frame positive contribution $\gamma^+$, and downward bars show the magnitude of the per-frame negative contribution $\gamma^-$ (left axis). The line plots the mean signed $\Delta$IoU per RoI (right axis).

proposals, and for each proposal $i$ we (i) cache pre-completion signals, namely seven hand-crafted RoI attributes and a proposal embedding; (ii) extract local partial points and apply 3DMF for one-step completion; (iii) merge visible and completed points to refit the box; and (iv) compute the BEV-IoU gain

$$
\begin{aligned}
g_i &\triangleq \Delta\text{IoU}_i = \text{IoU}_{\text{BEV}}^{\text{post}} - \text{IoU}_{\text{BEV}}^{\text{pre}}, \\
\gamma^+ &\triangleq \sum_{i \in \text{frame}} \max(g_i, 0), \\
\gamma^- &\triangleq \sum_{i \in \text{frame}} \max(-g_i, 0).
\end{aligned}
\tag{56}
$$

The seven RoI attributes are: (1) Completion Residual Energy, (2) Detector Score, (3) Heatmap Confidence, (4) Points in RoI, (5) Range, (6) Pre-completion IoU, and (7) Predicted IoU.

Aggregating over frames, a greedy oracle that applies completion whenever $g_i > 0$ attains a mean per-frame sum of positive BEV-IoU gains of $7.7 \times 10^{-3}$ IoU/frame.

**Gain Distribution Across Proposals.** To analyze how completion affects different proposal types, we report both per-frame statistics (sum of positive and negative $g_i$) and per-RoI statistics over range and point count, as shown in Figure 7. Without any selection, negative gains dominate. By range, the per-frame cumulative negative contribution is largest at mid-range distances (roughly 20–50 m), primarily because these bins contain many proposals, so small per-RoI degradations accumulate. By point count, the largest cumulative negatives appear below 32 points: although these RoIs are fewer, the underlying sparsity makes completion outcomes less certain, yielding larger downward sums.

Looking at the per-RoI mean, we observe a clear monotonic pattern. Nearer, denser RoIs show larger negative changes, whereas farther, sparser RoIs are less negative (closer to zero). This behavior matches intuition. When geometry is already well observed, completion can introduce small perturbations. When observations are sparse, completion tends to reduce under-specification, which lowers the magnitude of the negative drift even though the mean remains below zero.

**B.2. Completion Impact Dataset (CID)**

CID is constructed at the proposal level. For each RoI $i$, we define the one-step completion effect as $g_i$. All scalar attributes are standardized using statistics from the training split:

$$
z_k = \frac{u_k - \mu_k}{\sigma_k},
\tag{57}
$$

where $u_k$ is the raw value and $(\mu_k, \sigma_k)$ are the mean and standard deviation.

**Detector Score (Rectified) and Its Logit.** Let $s_{\text{cls}} \in (0, 1)$ denote the class score, and let $\hat{m}_0 \in [0, 1]$ denote the clipped IoU prediction obtained from the raw value $m$ via $\hat{m}_0 = \min\{1, \max\{0, (m + 1)/2\}\}$). With per-class weight $\lambda \in [0, 1]$,

$$s_{\text{det}} = s_{\text{cls}}^{1-\lambda} \hat{m}_0^{\lambda}, \qquad \ell_{\text{det}} = \log \frac{\tilde{s}_{\text{det}}}{1 - \tilde{s}_{\text{det}}}, \qquad \tilde{s}_{\text{det}} = \min\{1 - \varepsilon, \max\{\varepsilon, s_{\text{det}}\}\}, \tag{58}$$

with $\varepsilon = 10^{-6}$, and we standardize via $z_{\text{ldet}} = (\ell_{\text{det}} - \mu_{\text{ldet}})/\sigma_{\text{ldet}}$.

**Heatmap Confidence.** Given the decoded heatmap score $h$,

$$z_{\text{hm}} = \frac{h - \mu_{\text{hm}}}{\sigma_{\text{hm}}}. \tag{59}$$

**Range and Point Count.** With box center $(x, y)$ and visible point count $n$,

$$\begin{aligned}
r &= \sqrt{x^2 + y^2}, \\
z_{\text{ldist}} &= \frac{\log(1 + r) - \mu_{\text{ldist}}}{\sigma_{\text{ldist}}}, \\
z_{\text{lnpts}} &= \frac{\log(1 + n) - \mu_{\text{lnpts}}}{\sigma_{\text{lnpts}}}.
\end{aligned} \tag{60}$$

**Pre-completion IoU.** Let $\text{IoU}_{\text{BEV}}^{\text{pre}} \in [0, 1]$ denote the BEV IoU of the proposal box before completion and box refitting (as used in $g_i$). We standardize it as

$$z_{\text{iou0}} = \frac{\text{IoU}_{\text{BEV}}^{\text{pre}} - \mu_{\text{iou0}}}{\sigma_{\text{iou0}}}. \tag{61}$$

**Predicted-IoU Baseline.** If an IoU head exists, we also standardize $\hat{m}_0$ (falling back to $s_{\text{det}}$ if it is absent):

$$z_{\text{m0hat}} = \frac{\hat{m}_0 - \mu_{\text{m0hat}}}{\sigma_{\text{m0hat}}}. \tag{62}$$

**Completion Residual Energy.** Let $X \in \mathbb{R}^{N \times 3}$ be the partial point set in the canonical frame and let $M \in \{0, 1\}^N$ be its visibility mask, where $M_i = 1$ indicates an observed point. We denote $\overline{M} = \mathbf{1} - M$ and broadcast $\overline{M}$ to match the $N \times 3$ tensor shape. Let $Z_{\text{vis}}$ be the visible-code embedding. With a data-dependent noise scale $\sigma$ and velocity field $u(\cdot)$, we define

$$\begin{aligned}
\tilde{X} &= X + \sigma \left(\mathcal{N}(0, I) \odot \overline{M}\right), \\
U &= u(\tilde{X}, Z_{\text{vis}}) \odot \overline{M}, \\
E_{\text{res}} &= \sqrt{\frac{\|U\|_F^2}{\|\overline{M}\|_1}}, \qquad z_{\text{eres}} = \frac{E_{\text{res}} - \mu_{\text{eres}}}{\sigma_{\text{eres}}}.
\end{aligned} \tag{63}$$

Here, $\mathcal{N}(0, I)$ is an i.i.d. standard Gaussian tensor with the same shape as $X$.

We compute $z_{\text{eres}}$ only during CID construction and ACG training. At inference, we drop this feature to avoid extra computation, and ACG uses only the remaining standardized attributes and the proposal embedding.

**Concatenated input to ACG.** Let $q \in \mathbb{R}^{d_q}$ denote the optional proposal embedding (we use $d_q = 128$), and let $[\cdot \mid \cdot]$ denote concatenation. During CID construction and ACG training, we use

$$x_{\text{train}} = \left[z_{\text{eres}}, z_{\text{ldet}}, z_{\text{hm}}, z_{\text{lnpts}}, z_{\text{ldist}}, z_{\text{iou0}}, z_{\text{m0hat}} \mid q\right]. \tag{64}$$

At inference, we drop the completion residual energy feature $z_{\text{eres}}$ and use

$$x_{\text{test}} = \left[z_{\text{ldet}}, z_{\text{hm}}, z_{\text{lnpts}}, z_{\text{ldist}}, z_{\text{iou0}}, z_{\text{m0hat}} \mid q\right]. \tag{65}$$

If the embedding is disabled, we drop $q$ accordingly.

**CID Collection and Processing Pipeline.** We run the detector backbone and decode pre-NMS proposals, then compute the rectified score $s_{\mathrm{det}}$ and retain at most $T$ proposals per frame after class filtering, where $T$ is a per-frame proposal cap used during CID collection. RoIs that have too few points, lie beyond a maximum range, or have too small an initial IoU are discarded based on configurable thresholds. For each retained box $b = (x, y, z, \ell, w, h, \psi)$, we crop points in the world coordinate frame inside $b$, translate them to the box center, and apply isotropic normalization with $\eta = \max\{\ell, w, h\}$ to obtain the canonical partial $X$ and the visibility mask $M$. We encode $X$ into $Z_{\mathrm{vis}}$, perform a one-step 3DMF update in the canonical frame, de-normalize the completed points back to world coordinates, and merge them with observed points. The post-completion box is refit via robust $5\% \sim 95\%$ quantiles along each local axes while preserving the yaw angle $\psi$, which yields $g_i$. In the same pass, we compute all seven attributes, accumulate $(\mu, \sigma)$ statistics on the training split, and write sharded records to disk containing $\{g_i, s_{\mathrm{det}}, h, r, n, \mathrm{IoU}_{\mathrm{BEV}}^{\mathrm{pre}}, \hat{m}_0, E_{\mathrm{res}}, q\}$ together with the refit annotations for reproducibility. At inference time, $E_{\mathrm{res}}$ is not computed and is omitted from the ACG input.

### B.3. Adaptive Completion Gate: Training and Inference Details

This subsection complements Section 4 by specifying how the Adaptive Completion Gate (ACG) is trained on CID and how it is used at inference.

**Detector-specific Training.** ACG is trained on proposals produced by a specific detector and is not assumed to transfer across detectors. In all experiments, we train a separate ACG for each detector using its own CID. PointPlug is detector-agnostic in the sense that the completion and box refitting operate only on the RoIs available at the insertion point, independent of detector internals.

**Inputs and Normalization.** ACG takes the standardized RoI attributes and an optional proposal embedding as input. We use the standardized features defined in Appendix B.2 and compute the normalization statistics on the CID training split. At inference, we drop the training-only Completion Residual Energy feature (see Appendix B.2) and feed the remaining standardized attributes together with the proposal embedding.

**Architecture and Gating Score.** ACG is a lightweight MLP with LayerNorm on the input, two hidden layers of widths $(128, 64)$ with ReLU activations, and two linear heads. Given an input feature vector $x_i$, the network outputs a scalar ranking score $s_i$ and a non-negative magnitude predictor $u_i$:

$$(s_i, u_i) = \mathrm{ACG}_\theta(x_i), \qquad \kappa_i = s_i + \alpha\,\mathrm{ReLU}(u_i), \tag{66}$$

where $\alpha$ is a scalar hyperparameter. The gating score $\kappa_i$ is used only to rank proposals within each frame.

**Training Objective on CID.** Let $g_i = \Delta\mathrm{IoU}_i$ denote the BEV-IoU gain computed by completing and refitting the $i$-th proposal during CID construction. Due to the heavy-tailed and highly imbalanced distribution of $g_i$, we train ACG primarily as a ranker using pairwise logistic losses on stratified gain buckets. Specifically, for a pair $(i, j)$ where $i$ is sampled from a higher-gain bucket than $j$, we minimize

$$\mathcal{L}_{\mathrm{rank}} = \mathbb{E}_{(i,j)}\,\mathrm{BCEWithLogits}(\kappa_i - \kappa_j, \, 1), \tag{67}$$

and we add an auxiliary regression term that encourages the magnitude head to match the positive part of the gain:

$$\mathcal{L}_{\mathrm{reg}} = \mathbb{E}_i\,\mathrm{SmoothL1}(\mathrm{ReLU}(u_i), \, \mathrm{ReLU}(g_i)). \tag{68}$$

The final objective is a weighted sum $\mathcal{L}_{\mathrm{ACG}} = \lambda_{\mathrm{rank}}\mathcal{L}_{\mathrm{rank}} + \lambda_{\mathrm{reg}}\mathcal{L}_{\mathrm{reg}}$.

**Inference: Top-$K$ Selection.** At inference, we evaluate ACG on the detector's candidate RoIs before post-processing and select the top-$K$ RoIs per frame by the gating score $\kappa_i$. Only these RoIs are passed through one-step completion and box refitting; all remaining RoIs bypass completion and follow the original detector post-processing. In our experiments, completion is applied only to the Car class, so the top-$K$ selection is performed among Car RoIs.

## C. Completion and Generation: Training Details and Additional Results

### C.1. Network Architecture and Conditioning

As in the main text, the state at time $t$ is represented by a point set $X_t \in \mathbb{R}^{N \times 3}$. The network directly predicts the interval-averaged velocity $\bar{v}_\theta(X_t; r, t \mid s)$ and uses it for one-step or few-step average-velocity transport. All pointwise mappings are permutation equivariant and all set summaries are permutation invariant, so relabeling points does not alter the semantics.

**Permutation-Invariant Encoder (Shape Prior).** We compute a global shape prior $s \in \mathbb{R}^{256}$ from the observed subset using a lightweight PointNeXt-style (Qian et al., 2022) encoder. Concretely, a stem of 1D convolutions (Conv→GroupNorm→GELU) produces 64-channel features, followed by three inverse-residual MLP blocks (each keeping the channel size at 64) and a channel expansion to 128 channels. A symmetric max-pooling over points yields a 128-dimensional descriptor, which is then projected to a 256-dimensional prior $s$ by a two-layer MLP. This pathway is permutation invariant by construction and operates on partial inputs without depending on point order.

**Pointwise Decoder (Average-Velocity Head).** The transport module is a pointwise decoder that predicts $\bar{v}_\theta$ at each point. Each layer is a gated Concat-Squash linear map modulated by a context vector that concatenates sinusoidal embeddings of the absolute time and the step size with the shape prior:

$$c_{t,r} = \left[\, \mathrm{PE}(t),\ \mathrm{PE}(t-r),\ s \,\right] \in \mathbb{R}^{2d_e + d_s}, \tag{69}$$

where $d_e{=}64$ is the time-embedding dimension and $d_s{=}256$ is the prior dimension. The decoder implements the set-equivariant mapping

$$\bar{v}_\theta(X_t; r, t \mid s) = f_\theta(X_t;\, c_t), \tag{70}$$

with a residual formulation internally. During completion, a binary mask $M \in \{0, 1\}^N$ prevents updates on observed points:

$$\bar{v}_\theta^{\mathrm{miss}} = (\mathbf{1} - M) \odot \bar{v}_\theta(X_t; r, t \mid s). \tag{71}$$

This mask gating is permutation compatible and preserves the set structure. For inference, an exponential-moving-average (EMA) copy of the decoder (with decay 0.9995) is used to reduce prediction jitter and to match the teacher-free one-step and few-step procedures in the paper.

**Dimensionalities and Blocks.** The encoder produces a 256-dimensional prior $s$. The decoder consumes the concatenated context $c_t$ and stacks Concat-Squash layers with widths $3 \to 128 \to 256 \to 512 \to 256 \to 128 \to 3$; all gates and biases are generated from $c_t$. The overall design keeps pointwise operations permutation equivariant and the global prior permutation invariant, in line with the set-based formulation used throughout 3D MeanFlow.

### C.2. Training Schedule and Hyperparameters

We train 3D MeanFlow using the objective derived in Equations (41) and (42); we do not restate it here. This subsection documents the schedule, sampling strategy, and hyperparameters required to reproduce the reported results.

**Probability path and target instantaneous velocity.** We use the standard linear path between ground-truth point sets and Gaussian noise. Let $X_0 \sim p_{\mathrm{data}}$ be a ground-truth point cloud and let $\epsilon \sim \mathcal{N}(0, I)$ with $\epsilon \in \mathbb{R}^{N \times 3}$. For $t \in [0, 1]$, we define the intermediate state as

$$X_t = (1 - t)\, X_0 + t\, \epsilon, \tag{72}$$

where $t = 0$ corresponds to the data end and $t = 1$ corresponds to the noise end. The sample-wise instantaneous velocity along this path has a closed form:

$$v(X_t, t) \triangleq \frac{\mathrm{d}X_t}{\mathrm{d}t} = \epsilon - X_0. \tag{73}$$

This closed-form $v(X_t, t)$ is the instantaneous-velocity target used in the velocity-consistency objective of the main text.

For completion, we use the same binary mask $M \in \{0, 1\}^N$ as in Equation (71), where $M_i = 1$ marks observed points. We exclude observed points from velocity supervision by masking the target:

$$v^{\mathrm{miss}}(X_t, t) = (\mathbf{1} - M) \odot v(X_t, t). \tag{74}$$

Accordingly, all per-point velocity losses are evaluated only on missing points, using the masked prediction $\bar{v}_\theta^{\text{miss}}$ from Equation (71).

**Time and Step Sampling.** For each training pair we sample two times $r \le t$ with $t, r \in [0, 1]$, following the convention of the main text in which $t$ denotes the current (noisier) time, $r$ the target (cleaner) time, and $(t - r)$ the step length.

We bias sampling toward high-noise states while still covering the interior:

$$t \sim \begin{cases} \mathcal{U}[0.95, 1.00], & \text{with probability } p_{\text{high}}, \\ \text{Beta}(2, 2), & \text{otherwise.} \end{cases} \tag{75}$$

Given $t$, we sample $r$ such that the step length $(t - r)$ has an upper bound that expands toward the data end:

$$r \sim \mathcal{U}[t - \Delta_{\max}(t), t], \qquad \Delta_{\max}(t) \propto (1 - t). \tag{76}$$

The active fraction of pairs with $r \ne t$ is linearly increased during a short warm-up phase and then held near a target level:

$$\alpha_{\text{act}}(e) = \min\left(1, \frac{e}{E_{\text{warm}}}\right) p_{\text{mix}}, \tag{77}$$

where $\alpha_{\text{act}}(e)$ denotes the fraction of training pairs with $r < t$ (i.e., non-degenerate step length) at epoch $e$, with $E_{\text{warm}} \approx 10$ epochs and $p_{\text{mix}} \approx 0.4$. We optionally enforce a mild lower bound based on alignment diagnostics to avoid under-correction early in training.

**Jacobian–vector product for the total derivative.** The velocity-consistency target requires the total derivative

$$D_t[\bar{v}_\theta] = \underbrace{\partial_t \bar{v}_\theta}_{\text{time term}} + \underbrace{(\nabla_{X_t} \bar{v}_\theta) \, v(X_t, t)}_{\text{state / Jacobian term}}. \tag{78}$$

We compute $D_t[\bar{v}_\theta]$ as a single Jacobian–vector product with PyTorch autograd, without explicitly forming the Jacobian, by applying `jvp` to the (masked) average-velocity network $\bar{v}_\theta(X_t; r, t \mid s)$ with tangents $(v^{\text{miss}}(X_t, t), 0, 1)$ along the inputs $(X_t, r, t)$, respectively. The state tangent realizes the Jacobian term $(\nabla_{X_t} \bar{v}_\theta) \, v(X_t, t)$; the $r$ tangent is set to 0 since $r$ does not enter $D_t[\bar{v}_\theta]$; the $t$ tangent is set to 1, yielding the time term $\partial_t \bar{v}_\theta$. Their sum recovers $D_t[\bar{v}_\theta]$ in a single forward pass. The shape prior $s$ is treated as a constant in the JVP and therefore receives a zero tangent in the implementation. Algorithm 1 summarizes the full procedure.

**Pseudocode.** Algorithm 1 summarizes one training step in PyTorch-like pseudocode. The single `jvp` call with tangents $(v^{\text{miss}}, 0, 1)$ along inputs $(X_t, r, t)$ returns both $\bar{v}_\theta$ and the total derivative $D_t[\bar{v}_\theta]$ in one forward pass, exactly realizing the time-term/Jacobian-term decomposition in Equation (78). The construction reduces to standard flow matching training when $r = t$ (in which case the second term in the target vanishes), and otherwise implements the average-velocity matching that enables one-step inference.

**Optimization and Data.** Unless stated otherwise, we train on ShapeNet (Chang et al., 2015) with $N{=}2048$ output points and dataset-level mean and standard deviation normalization. We optimize with Adam (learning rate $1 \times 10^{-4}$, $\beta_1{=}0.5$, weight decay 0), an exponential learning-rate decay factor of 0.998, gradient clipping at 1.0, batch size 64, and 2500 training epochs. The decoder EMA decay is set to 0.9995. Time embeddings use $d_e{=}64$, and the shape prior has $d_s{=}256$. During training we monitor the cosine alignment between $\bar{v}_\theta$ and $v_t$, as well as the norm ratio $\|\bar{v}_\theta\|/\|v_t\|$ across time bins, and adjust a lower bound on $\alpha_{\text{act}}$ when necessary. All losses are permutation invariant and compatible with masking and the JVP-based total-derivative computation.

### C.3. Timing under Matched Hardware

The sampling-time entries in Tables 1 and 2 are cross-referenced from the corresponding papers, following the convention of prior work (Wu et al., 2023c). For a strict apple-to-apple comparison, we rerun all methods on the same machine with identical settings: a single NVIDIA A800 GPU, batch size 1, and $N{=}2048$ output points.

As shown in Table 8, 3DMF-1NFE remains the fastest method by a large margin under matched hardware, and 3DMF-5NFE is still roughly $5\times$ faster than PSF. The relative ordering and the order-of-magnitude speedup of 3DMF persist, confirming the efficiency advantage reported in the main text.

**Algorithm 1** 3DMF: One-step training procedure.

```
# fn(X, r, t, s): masked average-velocity network
# X0: ground-truth shape; M: observation mask; lam: shape-loss weight

t, r     = sample_t_r()                        # time sampling
eps      = randn_like(X0)
s        = encoder(M * X0)

Xt       = (1 - t) * X0 + t * eps              # linear path
v        = eps - X0                            # instantaneous velocity
v_miss   = (1 - M) * v                         # mask out observed pts

vbar, Dvbar = jvp(fn, (Xt, r, t, s), (v_miss, 0, 1, 0))
vbar_tgt    = v - (t - r) * Dvbar
loss_v      = metric(vbar - stopgrad(vbar_tgt)) # velocity consistency

Xr_hat = Xt - (t - r) * vbar                   # one-step prediction
loss_s = shape_loss(Xr_hat)                    # shape-level constraint

loss = loss_v + lam * loss_s
```

*Table 8.* Per-shape sampling time under matched hardware.

| Model | PointFlow | PVD | DPF-Net | PSF | 3DMF-1NFE | 3DMF-5NFE |
|---|---|---|---|---|---|---|
| Time (s) ↓ | 0.184 | 20.37 | 0.227 | 0.061 | **0.005** | 0.013 |

## C.4. Model Footprint and Inference Throughput

To address concerns about footprint, we report parameter count, peak GPU memory, and throughput of 3DMF under a fixed inference protocol (batch=1, $N$=2048, NVIDIA A800). Many prior works do not report these statistics, so we provide ours for transparency in Table 9.

## C.5. Additional Completion Results

Under the same evaluation protocol and implementation settings as in the main paper, we additionally report the 2NFE configuration and provide broader qualitative comparisons across categories. Quantitative results are restricted to the three standard ShapeNet classes (Airplane, Chair, and Car), as summarized in Table 10. For the remaining categories that are not covered by the main quantitative table, we present representative completion visualizations without reporting numerical scores; see Figure 9. Overall, 2NFE narrows the accuracy gap to 5NFE while essentially preserving millisecond-level sampling latency. Compared to 1NFE, it more reliably reconstructs thin structures and high-curvature regions while maintaining stable global topology, yielding a favorable balance between quality and efficiency.

## C.6. Additional Generation Results

To further assess our one-step and few-step paradigm in unconditional generation, we compare against additional representative methods under identical data and timing setups, and expand qualitative results to more categories. The quantitative comparison remains limited to Airplane, Chair, and Car; Table 11 reports 1-NNA (computed with CD and EMD) together with per-shape sampling time. We additionally report distribution-level metrics MMD↓ and COV(%)↑ in Table 12. Consistent with the main text, 3DMF-1NFE attains the fastest sampling, 2NFE improves 1-NNA with modest overhead, and 5NFE offers additional detail and completeness in several settings. For categories beyond Airplane, Chair, and Car, we provide visualizations to illustrate structural plausibility and distributional coverage, without additional quantitative metrics; see Figure 11.

*Table 9.* Compute and footprint summary for 3DMF-1NFE completion under a fixed protocol ($B{=}1$, $N{=}2048$).

| Item | Value |
|---|---|
| Model size (parameters) | 2.8M |
| FLOPs per forward (1-step) | $7.994 \times 10^8$ (0.799 GFLOPs) |
| Peak GPU memory (allocated / reserved) | 50.6 MiB / 62.0 MiB |
| Training set size | 2458 |
| Number of epochs | 2500 |
| Batch size | 64 |
| Total training FLOPs (est., $\times 3$ for FW/BW/update) | $1.474 \times 10^{16}$ (14.737 PFLOPs) |
| Training time (reported) | 8.5 hours |
| Test set size | 352 |
| Total inference FLOPs (test set) | $2.814 \times 10^{11}$ |
| Inference FLOPs per 1000 instances | $7.994 \times 10^{11}$ |
| FLOPs tool | fvcore |

*Table 10.* Additional completion results on Airplane, Chair, Car compared with baselines.

| Model | Airplane CD↓ | Airplane EMD↓ | Chair CD↓ | Chair EMD↓ | Car CD↓ | Car EMD↓ | Sampling Time (s) |
|---|---|---|---|---|---|---|---|
| PointFlow (Yang et al., 2019) | 0.403 | 1.180 | 2.707 | 3.649 | 1.803 | 2.851 | 0.27 |
| SoftFlow (Kim et al., 2020) | 0.404 | 1.198 | 2.786 | 3.295 | 1.850 | 2.789 | 0.12 |
| PVD (Zhou et al., 2021) | 0.442 | 1.030 | 3.211 | 2.939 | 1.774 | 2.146 | 29.9 |
| DPF-Net (Shuai et al., 2023) | 0.528 | 1.105 | 2.763 | 3.320 | 1.396 | 2.318 | 0.34 |
| PSF (Wu et al., 2023c) | - | **1.004** | - | 2.937 | - | 2.194 | 0.04 |
| PointNSP (Meng et al., 2025) | 0.401 | 1.008 | 2.702 | **2.878** | 1.384 | 2.068 | 5.48 |
| **3DMF-1NFE** *(ours)* | 0.639 | 1.167 | 2.933 | 3.541 | 1.224 | 1.896 | **0.005** |
| **3DMF-2NFE** *(ours)* | 0.443 | 1.139 | 2.512 | 3.537 | 0.982 | 1.773 | 0.007 |
| **3DMF-5NFE** *(ours)* | **0.369** | 1.121 | **2.138** | 3.188 | **0.975** | **1.752** | 0.013 |

# D. PointPlug: Proposal Dependency Analysis and Extended Detection Results

### D.1. Proposal Dependency Analysis

This subsection complements the PointPlug description in Section 4 by quantifying how proposal dependency affects the cases that PointPlug can improve. We use FSHNet as a strong LiDAR detector and report results on the widely used nuScenes benchmark.

PointPlug is inserted after the detector head and can only operate on the candidate RoIs produced at that stage. It refines these RoIs through completion-driven box and score updates, and it cannot help a false negative when no candidate RoI is produced near the object.

We quantify this dependency by measuring *proposal coverage* for ground-truth cars at the PointPlug input. We use the detector's candidate RoIs before post-processing, prior to score filtering and NMS. For each ground-truth car box, we check whether at least one candidate RoI has its box center within a distance $d$ from the ground-truth center. We report results for $d = 2$ m and $d = 4$ m, and we stratify by range.

Table 14 shows that within the official 0–50 m evaluation range, candidate RoIs are almost always present near ground-truth cars. More importantly, a substantial fraction of false negatives in this range still have at least one nearby candidate RoI at the PointPlug input. This indicates that many misses arise because the candidate exists but is later discarded or poorly localized, rather than because no hypothesis was produced at all. PointPlug targets these cases by refining boxes and confidence scores before post-processing. Beyond 50 m, coverage for false negatives drops sharply, so most long-range misses have no candidate RoI to refine, which is a fundamental limitation of proposal-refinement designs. Nevertheless, for the far-range objects that do have candidate RoIs, refinement is often most beneficial, which explains why we observe the largest relative gains at 50 m+.

*Table 11.* Additional generation results on Airplane, Chair, Car compared with baselines.

| Model | Airplane | | Chair | | Car | | Sampling Time (s) |
|---|---|---|---|---|---|---|---|
| | CD↓ | EMD↓ | CD↓ | EMD↓ | CD↓ | EMD↓ | |
| 1-GAN (Achlioptas et al., 2018) | 87.30 | 93.95 | 68.58 | 83.84 | 66.49 | 88.78 | 0.03 |
| PointFlow (Yang et al., 2019) | 75.68 | 70.74 | 62.84 | 60.57 | 58.10 | 56.25 | 0.27 |
| SoftFlow (Kim et al., 2020) | 76.05 | 65.80 | 59.21 | 60.05 | 64.77 | 60.09 | 0.12 |
| ShapeGF (Cai et al., 2020) | 80.00 | 76.17 | 68.96 | 65.48 | 63.20 | 56.53 | 0.34 |
| SetVAE (Kim et al., 2021) | 75.31 | 77.65 | 58.76 | 61.48 | 59.66 | 61.48 | 0.03 |
| PVD (Zhou et al., 2021) | 73.82 | 64.81 | 56.26 | 53.32 | 54.55 | 53.83 | 29.9 |
| LION (Vahdat et al., 2022) | 72.99 | 64.21 | 55.67 | 53.82 | 53.47 | 53.21 | 27.1 |
| DPF-Net (Shuai et al., 2023) | 75.18 | 65.55 | 62.00 | 58.53 | 62.35 | 54.48 | 0.34 |
| PSF (Wu et al., 2023c) | **71.11** | **61.09** | 58.92 | 54.45 | 57.19 | 56.07 | 0.04 |
| Tiger (Ren et al., 2024) | 73.02 | 64.10 | 55.15 | 53.18 | 53.21 | 53.95 | 9.73 |
| PointNSP (Meng et al., 2025) | 72.24 | 63.69 | **54.54** | **52.85** | 52.17 | 51.85 | 5.48 |
| **3DMF-1NFE** *(ours)* | 76.19 | 70.27 | 58.73 | 55.01 | 54.03 | 53.24 | **0.005** |
| **3DMF-2NFE** *(ours)* | 73.99 | 65.81 | 57.17 | 54.12 | 52.60 | 52.45 | 0.007 |
| **3DMF-5NFE** *(ours)* | 73.38 | 64.32 | 56.55 | 53.70 | **52.06** | **51.81** | 0.013 |

*Table 12.* Additional unconditional generation results on ShapeNet using MMD↓ and COV(%)↑ computed under CD / EMD distances.

| Model | Airplane | | | | Chair | | | | Car | | | |
|---|---|---|---|---|---|---|---|---|---|---|---|---|
| | MMD ↓ | | COV(%) ↑ | | MMD ↓ | | COV(%) ↑ | | MMD ↓ | | COV(%) ↑ | |
| | CD | EMD | CD | EMD | CD | EMD | CD | EMD | CD | EMD | CD | EMD |
| 1-GAN (2018) | 0.340 | 0.583 | 38.52 | 21.23 | 2.589 | 2.007 | 41.99 | 29.31 | 1.532 | 1.226 | 38.92 | 23.58 |
| PointFlow (2019) | 0.224 | 0.390 | 47.90 | 46.41 | **2.409** | 1.595 | 42.90 | 50.00 | **0.901** | 0.807 | 46.88 | 50.00 |
| SoftFlow (2020) | 0.231 | 0.375 | 46.91 | 47.90 | 2.528 | 1.682 | 41.39 | 47.43 | 1.187 | 0.859 | 42.90 | 44.60 |
| ShapeGF (2020) | 2.703 | 0.659 | 40.74 | 40.49 | 2.889 | 1.702 | 46.67 | 48.03 | 9.232 | **0.756** | **49.43** | 50.28 |
| PVD (2021) | 0.224 | 0.380 | **48.88** | 52.09 | 2.622 | 1.556 | **49.84** | **50.60** | 1.077 | 0.794 | 41.19 | 50.56 |
| DPF-Net (2023) | 0.264 | 0.409 | 46.17 | 48.89 | 2.536 | 1.632 | 44.71 | 48.79 | 1.129 | 0.853 | 45.74 | 49.43 |
| PSF (2023c) | 0.221 | **0.366** | 46.17 | 52.59 | 2.624 | 1.573 | 46.71 | 49.84 | 1.023 | 0.802 | 42.89 | 53.12 |
| **3DMF-1NFE** *(ours)* | 0.236 | 0.398 | 44.87 | 49.82 | 2.551 | 1.617 | 45.13 | 48.82 | 1.069 | 0.807 | 41.27 | 51.81 |
| **3DMF-2NFE** *(ours)* | 0.223 | 0.391 | 45.10 | 50.49 | 2.499 | 1.586 | 46.01 | 49.30 | 1.021 | 0.779 | 42.51 | 53.09 |
| **3DMF-5NFE** *(ours)* | **0.219** | 0.385 | 45.74 | 50.64 | 2.491 | **1.554** | 46.32 | 49.51 | 1.015 | 0.776 | 42.50 | **53.26** |

## D.2. Extended 3D Object Detection Results with PointPlug

This section reports full nuScenes metrics for detectors equipped with *PointPlug*, complementing the aggregate results in the main text (see Table 13). The evaluation follows the same protocol and settings as in the main text, using the official nuScenes metrics (NDS, mAP) and single-class completion on Car. Because only the Car class is completed, changes in NDS and mAP are modest and primarily driven by improvements on Car; results for the remaining classes are shown for completeness.

**Range scope and instance counts.** We further break down Car performance by range in the main paper (Table 3). The official nuScenes metrics consider only objects within 50m, so cars beyond 50m do not contribute to NDS/mAP. On the nuScenes *val* set, this corresponds to 31,338 Car instances in 0–30m and 18,993 in 30–50m, while 9,155 instances at 50m+ are excluded from the official aggregates.

## E. Additional Ablations and Analyses

### E.1. Design Ablations of 3DMF

In this section, we analyze ablations of 3DMF that target design factors closely tied to one-step and few-step behavior. For the *Car* completion task, quantitative results for the three core choices—the fraction of pairs with $r \neq t$, the positional embedding of $(t, r)$, and the JVP tangent—are summarized in Table 15.

**Ratio of Sampling with $r \neq t$.** Table 15a varies the proportion of training pairs with $r \neq t$. A zero ratio (reducing to standard Flow Matching) fails to produce meaningful 1NFE behavior, whereas introducing a nonzero fraction enables

*Table 13.* NuScenes *val* set, class-wise detection results with PointPlug (*-pp*). Only the *Car* class is completed; all classes are shown for completeness. 'T.L.', 'C.V.', 'Ped.', 'M.T.', 'T.C.', and 'B.R.' denote trailer, construction vehicle, pedestrian, motor, traffic cone, and barrier, respectively.

| Model | NDS | mAP | Car | Truck | Bus | T.L. | C.V. | Ped. | M.T. | Bike | T.C. | B.R. |
|---|---|---|---|---|---|---|---|---|---|---|---|---|
| VoxelNeXt (2023) | 68.7 | 63.5 | 83.9 | 55.5 | 70.5 | 38.1 | 21.1 | 84.6 | 62.8 | 50.0 | 69.4 | 69.4 |
| **VoxelNeXt-pp** | 68.9 | 64.0 | 84.3 | 55.4 | 70.4 | 38.1 | 21.1 | 84.4 | 62.8 | 49.9 | 69.4 | 69.3 |
| FSHNet (2025) | 71.7 | 68.1 | 88.7 | 61.4 | 79.3 | 47.8 | 26.3 | 89.3 | 76.7 | 60.5 | 78.6 | 72.3 |
| **FSHNet-pp** | 71.8 | 68.5 | 89.3 | 61.4 | 79.2 | 47.8 | 26.3 | 89.3 | 76.7 | 60.5 | 78.6 | 72.4 |

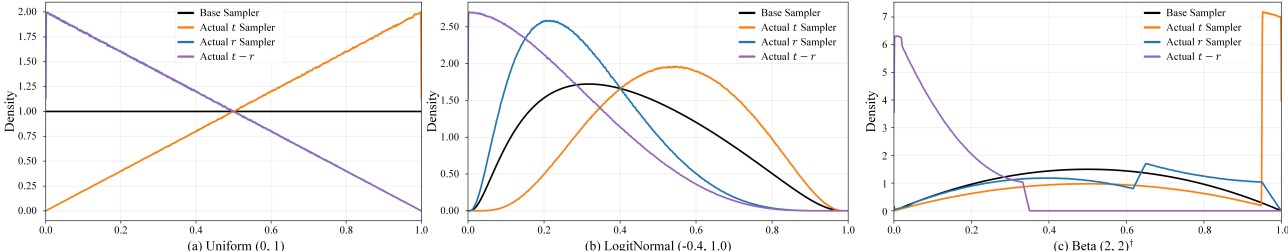

*Figure 8.* Induced time-sampling distributions under different base samplers. (a) $\mathrm{Uniform}(0,1)$; (b) $\mathrm{LogitNormal}(-0.4,1.0)$; (c) the high-$t$ $\mathrm{Beta}(2,2)^{\dagger}$ sampler used in 3DMF. In each panel, the black curve shows the base sampler, and the orange, blue, and purple curves show the empirical densities of $t$, $r$, and $t - r$ after enforcing $t \geq r$.

effective one-step generation. In our setting, a moderate ratio yields the best trade-off, while pushing the ratio too high degrades performance. This trend mirrors the MeanFlow (Geng et al., 2025a) observation that a nonzero $r \neq t$ ratio is necessary for 1NFE and that the optimal mixture is moderate.

**Positional Embedding of $(t, r)$.** Table 15b compares several conditioning choices for encoding time. Encoding time using $(t, t - r)$ attains the strongest results, with $(t, r)$, performing closely and even using only the interval $t - r$ remaining viable. This ordering is consistent with the MeanFlow study on one-step generation, which found that $(t, t - r)$ is slightly preferred but all three variants are reasonable.

**JVP Computation.** Table 15c evaluates different Jacobian–vector product (JVP) tangents used in the 3DMF identity. Only the correct tangent $(v, 0, 1)$ yields reliable 1NFE performance; using $(v, 0, 0)$, $(v, 1, 0)$, or $(v, 1, 1)$ substantially degrades results. This matches the MeanFlow analysis that the total derivative couples $[\partial_z u, \partial_r u, \partial_t u]$ with the tangent $[v, 0, 1]$, and that incorrect JVP computation breaks one-step generation.

**Effect of the Number of Update Steps.** We vary the number of 3DMF updates and report quality and latency in Table 16. Quality improves markedly from 1NFE to 2NFE and peaks at 5NFE (best CD and EMD). Beyond five steps, both CD and EMD deteriorate, with pronounced degradation at 20–100 steps. Latency grows roughly linearly with the step count: 1NFE is about 5 ms, 5NFE is about 13 ms, and 10NFE is about 27 ms per shape. We attribute the initial gains (1–5 steps) to additional correction opportunities that reduce one-step transport errors on fine-scale details, while the degradation at longer horizons is consistent with error accumulation and distribution shift in intermediate states under repeated updates.

**Distributional Effects of the Swap Constraint.** When $t$ and $r$ are sampled independently from the same base distribution and then swapped to enforce $t \geq r$, the induced marginals no longer follow the base distribution. Figure 8 (a) and (b) plot the resulting densities of $t$, $r$, and $t - r$ after swapping. With a uniform base distribution, the distribution of $t - r$ matches that of $r$, so the two curves coincide. With a logit-normal base $\mathrm{LogitNormal}(-0.4, 1.0)$, swapping pushes $t$ toward larger values and $r$ toward smaller values, causing $t - r$ to concentrate near zero. In contrast, our sampler draws $t$ first with explicit emphasis on high-noise times, then samples $r$ from a $t$-dependent interval whose width scales with $t$; this coupling controls the update horizon. Although training does not explicitly include the endpoint $t - r = 1$, the learned average-velocity field extrapolates well to this endpoint in our experiments, enabling stable one-step generation.

*Table 14.* Proposal coverage for ground-truth cars at the PointPlug input on nuScenes *val* with FSHNet. All-GT averages over all cars, while FN-only averages over cars that are false negatives under the official evaluation. GT and FN denote the number of ground-truth cars and false negatives in each range bin.

| Range (m) | All-GT (%) | | FN-only (%) | | GT | FN |
|---|---|---|---|---|---|---|
| | $d=2$ | $d=4$ | $d=2$ | $d=4$ | | |
| 0 – 30 | 99.68 | 99.80 | 15.13 | 47.90 | 31,338 | 119 |
| 30 – 50 | 98.62 | 98.97 | 22.42 | 42.48 | 18,993 | 339 |
| 0 – 50 | 99.28 | 99.49 | 20.52 | 43.89 | 50,331 | 458 |
| 50 + | 40.55 | 42.27 | 1.04 | 3.91 | 9,155 | 5,500 |

*Table 15.* Ablation study on 3DMF time-sampling design for the Car completion task. CD and EMD are reported for the 1NFE and 5NFE settings.

| % of $r \neq t$ | 1NFE | | 5NFE | |
|---|---|---|---|---|
| | CD | EMD | CD | EMD |
| 0% | 4.006 | 5.334 | 2.309 | 2.237 |
| 20% | 1.224 | 1.896 | 0.975 | 1.752 |
| 40% | 1.219 | 2.169 | 0.998 | 1.813 |
| 100% | 1.350 | 2.642 | 1.191 | 2.045 |

*(a)* Ratio of sampling with $r \neq t$.

| Pos. Embed | 1NFE | | 5NFE | |
|---|---|---|---|---|
| | CD | EMD | CD | EMD |
| $t-r$ only | 1.568 | 2.351 | 1.182 | 2.057 |
| $(t,r)$ | 1.320 | 2.123 | 1.037 | 1.861 |
| $(t,t-r)$ | 1.224 | 1.896 | 0.975 | 1.752 |
| $(t,r,t-r)$ | 1.265 | 1.980 | 0.994 | 1.784 |

*(b)* Positional embedding of $(t,r)$.

| JVP Tangent | 1NFE | | 5NFE | |
|---|---|---|---|---|
| | CD | EMD | CD | EMD |
| $(v,0,0)$ | 2.025 | 3.089 | 1.540 | 2.336 |
| $(v,0,1)$ | 1.224 | 1.896 | 0.975 | 1.752 |
| $(v,1,0)$ | 2.213 | 3.559 | 1.722 | 2.523 |
| $(v,1,1)$ | 1.431 | 2.226 | 1.127 | 1.956 |

*(c)* JVP tangents used in the 3DMF identity.

## E.2. Permutation Equivariance

Point clouds are unordered sets without a canonical indexing. Prior work on set-structured learning has shown that order-sensitive architectures can exhibit significant dependence on arbitrary orderings, motivating permutation-invariant/equivariant designs and robustness checks under reindexing perturbations (Vinyals et al., 2015; Murphy et al., 2018). We therefore conduct an ordering mismatch stress test to validate that permutation equivariance is practically important for consistent 1NFE inference on point sets.

**Protocol.** During training, we impose a fixed deterministic ordering on every input point cloud by sorting points lexicographically by $(x, y, z)$. At test time, we evaluate two settings on the same test set. In the *matched* setting, we apply the same ordering rule as in training. In the *mismatched* setting, we either apply a different deterministic ordering (e.g., lexicographic sorting by $(y, x, z)$) or randomly permute the input points for each forward pass. Importantly, the metric itself is unchanged: the reported CD is always computed between the predicted output point cloud and the ground truth.

**Models.** We compare our permutation-equivariant 3DMF with an order-sensitive ablation based on the same PointNeXt encoder. The ablation minimally breaks permutation symmetry by adding a learnable absolute index embedding $e_i$ to each point feature according to its position $i$ in the ordered sequence. All other components are kept identical.

As shown in Table 17, 3DMF remains nearly invariant to reindexing, while the order-sensitive variant suffers a large and highly variable degradation, indicating that enforcing permutation equivariance is critical for robust 1NFE inference on point sets.

## F. Additional Qualitative Results

*Table 16.* Ablation study on the number of update steps for point cloud completion.

| Step | CD↓ | EMD↓ | Sampling Time (ms) |
|------|------|------|------|
| 1 | 1.224 | 1.896 | 5 |
| 2 | 0.983 | 1.771 | 7 |
| 5 | 0.975 | 1.752 | 13 |
| 10 | 0.982 | 1.920 | 27 |
| 20 | 1.028 | 2.341 | 46 |
| 50 | 1.171 | 2.891 | 100 |
| 100 | 1.235 | 3.064 | 189 |

*Table 17.* Ordering-mismatch stress test. Only the test-time input ordering changes. Mismatch results are mean±std over 10 random permutations per test input; ∆CD denotes the increase of the mean mismatched CD over the matched CD.

| Method | CD (matched)↓ | CD (mismatched)↓ | ∆CD↓ |
|--------|------|------|------|
| 3DMF-1NFE | 1.224 | 1.225±0.002 | 0.001 |
| Order-sensitive | 1.276 | 1.640±0.429 | 0.364 |

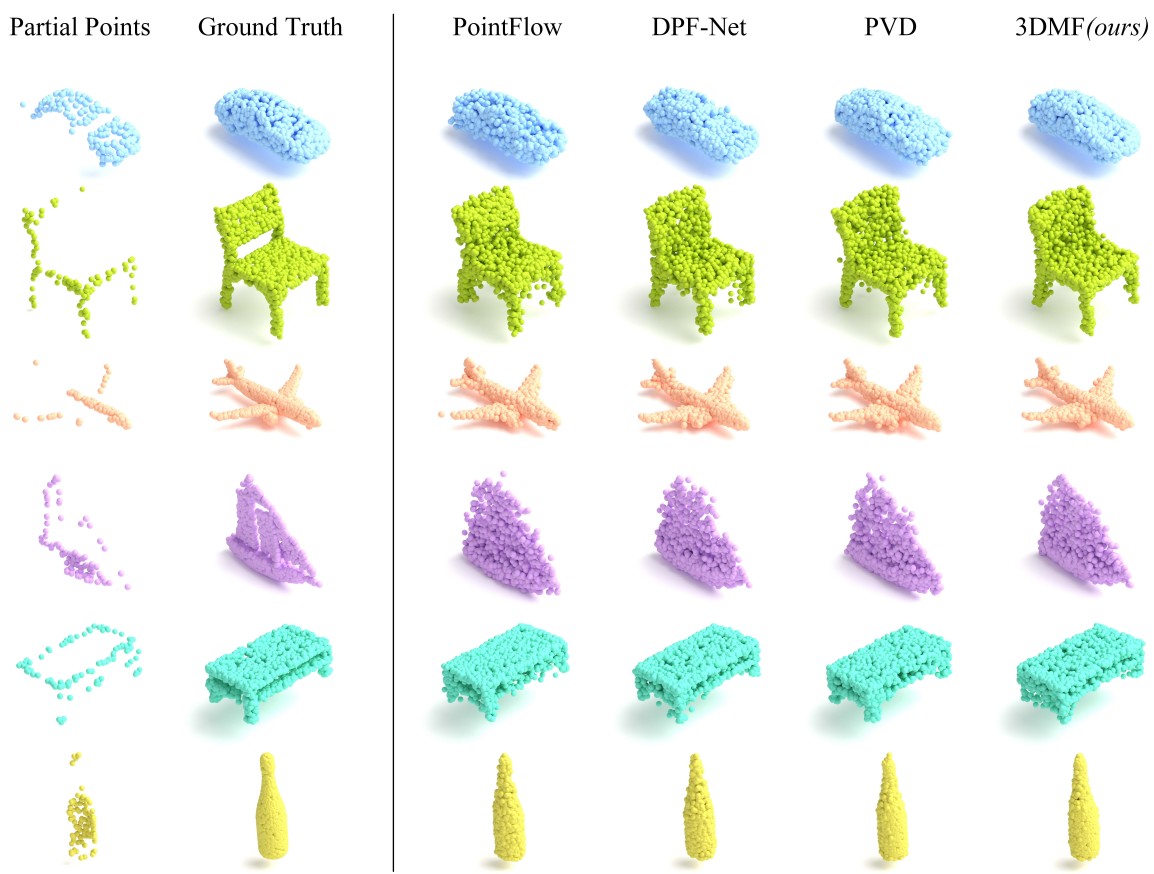

*Figure 9.* Additional point cloud completion comparisons.

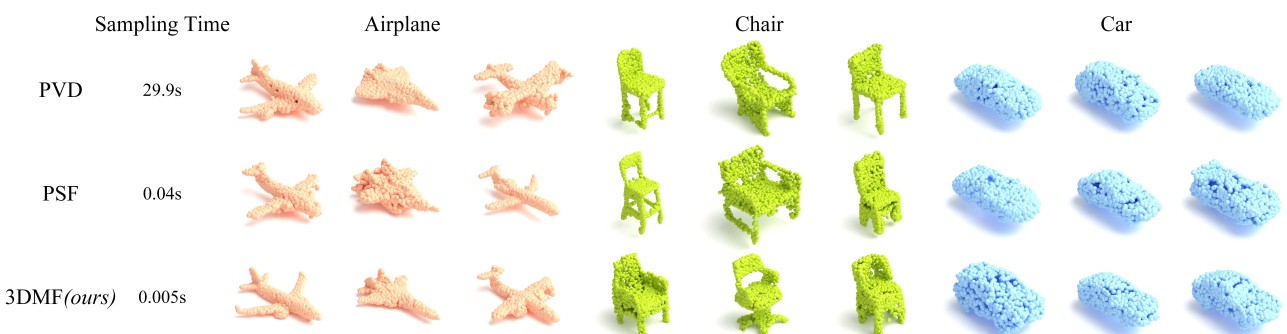

*Figure 10.* Point cloud generation comparison. 3DMF is shown in the one-step setting (3DMF-1NFE). Per-shape sampling times are listed on the left. 3DMF attains competitive visual quality while operating at the fastest sampling rate.

Airplane

Chair

Car

Vessel

Bookshelf

Table

Bottle

Bowl

*Figure 11.* Additional qualitative unconditional generation results on more ShapeNet categories (per-category training).

