# OpenReview forum: "3D MeanFlow: One-Step Point Cloud Completion and Generation via Average-Velocity Transport"
_ICML.cc/2026/Conference — ICML 2026 regular_

### Official Review · Reviewer_Pt1P · 2026-02-27

**Soundness:** 3
**Presentation:** 2
**Significance:** 2
**Originality:** 2
**Overall Recommendation:** 4
**Confidence:** 4

**Summary:**

This paper proposes **3D MeanFlow (3DMF)**, a teacher-free one-step generative model for **point cloud completion** and **unconditional generation**. Building on MeanFlow, the method learns an **interval-averaged velocity field** in a **permutation-equivariant** set formulation and trains it with an instantaneous-vs-average consistency loss plus a **Chamfer-based shape-level regularizer** for stability. The paper also introduces **PointPlug**, which uses an adaptive gate to apply one-step completion to a small number of detector RoIs, improving nuScenes and KITTI detection with small latency overhead.

**Compliance With Llm Reviewing Policy:**

Affirmed.

**Final Justification:**

This paper explores a practical problem, and the authors' responses clarified several design choices and experimental details, which strengthened my confidence in the paper's rationale and clarity. However, my main concerns have not been entirely eliminated: the methodological innovation remains limited, and the current empirical support is insufficient to fully demonstrate the paper's broader significance. Therefore, my overall rating has increased by one point from before, now at weak acceptance.

**Key Questions For Authors:**

1. **Training data for detection completion:** Is the 3DMF model used in nuScenes/KITTI trained purely on ShapeNet Car, or is there any fine-tuning on real LiDAR RoIs (even self-supervised)? How sensitive are the detection gains to this choice?
2. **Endpoint extrapolation for one-step:** How often does training sample large step sizes close to \(t-r=1\)? Do you have an ablation where you explicitly include endpoint-like pairs (or a curriculum that increases step size) and measure the effect on 1NFE fidelity?
3. **Timing fairness:** Are the baseline sampling times measured with your own implementation under the same device/precision/batch settings? If some are taken from prior papers, can you provide a normalized timing comparison under a single codebase/hardware configuration?
4. **ACG vs heuristics:** How does ACG compare to simple selection heuristics based on range, point count, or detector uncertainty? If ACG is substantially better, which input features contribute most (ablation on the RoI attributes and proposal embedding)?

**Limitations:**

yes

**Strengths And Weaknesses:**

**Strengths**
- The extension from MeanFlow to point sets is conceptually well-motivated: enforcing **permutation equivariance** for velocity fields and showing the identity remains well-defined for unordered points directly targets a key 3D point-cloud requirement (Eq. 4–6; App. A.1).
- The training objective is a clear adaptation of MeanFlow: a self-consistency target built from the instantaneous velocity along the linear path and a total-derivative term computed via **JVP**, plus a global **shape-level Chamfer** constraint that empirically stabilizes one-step updates (Eq. 9–13; Table 6).
- The paper includes useful ablations validating non-trivial design choices (time sampling in Table 5, fraction of pairs with \(r \neq t\) in Table 14a, and the correct JVP tangent choice in Table 14c).
- The ordering-mismatch stress test (Table 16) provides concrete evidence that permutation-equivariant modeling matters in practice, not only in principle.

**Weaknesses**
- The method’s theoretical grounding is still mostly **identity-based** rather than **distributional**: the MeanFlow identity is exact for the true average velocity along the true trajectory, but 3DMF learns an approximate v-bar field via a stop-gradient fixed-point style objective. The paper does not provide conditions under which minimizing the proposed loss yields a transport map that is consistent with the intended data-to-noise dynamics (especially for one-step inference).
- One-step sampling uses the endpoint interval \((r,t)=(0,1)\), but training samples \((r,t)\) from a designed distribution (App. C.2). The paper states the model extrapolates reliably to the endpoint, but a more explicit analysis of **how much endpoint-like supervision the model actually sees** would strengthen the core “one-step” claim.
- The reliance on JVP/total-derivative computation adds implementation sensitivity (masking, time/step conditioning, tangent correctness). The appendix helps, but a short algorithm box/pseudocode in the main paper would reduce reproduction risk for readers less familiar with JVP-based training.
- The few-step operator degrades beyond ~5 updates (Table 15), suggesting notable **distribution shift** under iteration. This limits the model’s use as a general multi-step solver and raises questions about intermediate-time calibration of the learned field.
- The shape-level Chamfer loss is effective but also known to allow density artifacts (e.g., clustering) and does not directly enforce uniform surface sampling; the paper does not quantify whether such artifacts increase under strong shape regularization.

---

### Presentation

**Strengths**
- The paper is well-structured and generally easy to follow: motivation (latency), method (set-based MeanFlow), then system integration (PointPlug), then experiments/ablations.
- The appendices contain substantial engineering detail (architecture, time sampling, CID/ACG construction), and even provide compute/footprint reporting (Table 8), which is appreciated.

**Weaknesses**
- The notation around time variables (tau vs t, r, delta, and step size) can be confusing. A short notation table and a single consistent naming scheme would improve readability.
- The main text does not fully connect the mathematical decomposition (time term vs state/Jacobian term) to the exact JVP computation. This connection is mostly in the appendix and ablations (Table 14c), but it is central to why training works.
- PointPlug introduces several design choices (CID filters, quantile-based box refitting, score fusion rule, and detector-specific ACG training). Many are only in the appendix; summarizing the “minimum necessary” implementation details in the main text would make the contribution easier to assess and reuse.

---

### Significance

**Strengths**
- Achieving millisecond-level point cloud completion with competitive fidelity is practically valuable for robotics and autonomous driving, where latency budgets are strict.
- The proposal-level integration (PointPlug) is a meaningful step toward measuring *downstream* utility of completion, and the range-stratified nuScenes analysis is informative (Table 3).

**Weaknesses**
- The **official** nuScenes metrics exclude objects beyond 50m, while the biggest relative gains are reported at 50m+. As a result, improvements in aggregated NDS/mAP are modest (Table 3, Table 12), which weakens the headline practical impact under standard evaluation.
- PointPlug is only evaluated with completion applied to the **Car** class, and the completion model is trained per ShapeNet category. The paper’s real-world significance would be stronger with evidence that the approach generalizes to additional classes and more varied real-LiDAR shape distributions.
- Training separate models per category and using long training schedules (e.g., 2500 epochs per category as reported) may limit accessibility for practitioners who need multi-class or open-world completion.

---

### Originality (novelty / contribution)

**Strengths**
- The main novelty is a careful adaptation of a modern one-step generative paradigm (MeanFlow) to **unordered 3D point sets**, including an explicit permutation-equivariant formulation and a lightweight global-shape stabilizer for one-step updates.
- PointPlug is a novel “systems” contribution: integrating one-step completion into detectors with an explicit top-K latency budget is a useful and non-obvious design.

**Weaknesses**
- The core generative objective is closely derived from MeanFlow, and several components (set encoder + pointwise decoder; Chamfer regularization; masking observed points) are standard in point-cloud generative modeling. The novelty is therefore more **in adaptation and stabilization** than in a fundamentally new modeling principle.
- The ACG gating mechanism requires detector- and dataset-specific CID construction and training, which reduces the “drop-in” nature of PointPlug in practice.
- Some aspects resemble existing proposal refinement patterns (rank proposals, refine a subset, update scores/boxes). The distinctive part is “completion-driven refinement,” but the paper could better isolate what is uniquely enabled by 3DMF vs what could be achieved by simpler geometric priors.

---

### Experimental strengths and weaknesses (and important missing experiments)

**Strengths**
- Strong speed/latency reporting: 3DMF-1NFE is ~5 ms per shape and 3DMF-5NFE ~13 ms on A800, with compute/footprint stats provided (Table 8).
- The ablation suite is thoughtful and targets key one-step failure modes (time sampling, lambda, ordering robustness, JVP tangents).

**Weaknesses / high-value additions**
1. **One-step quality is not uniformly competitive** in completion: for Airplane and Chair, 3DMF-1NFE is noticeably worse in CD/EMD than several baselines (Table 1/9). The best accuracy often comes from 5NFE, which partially shifts the contribution from “one-step” to “few-step but fast.”
2. Clarify **baseline timing methodology** (re-implemented vs cited; same hardware; same precision; batch size; use of compilation). Since speed is a central claim, consistent measurement matters.
3. Add a **sparsity sweep** for completion (vary number of observed points and/or view coverage). The current setup fixes 200 visible points, but real applications vary widely in point count and occlusion.
4. For PointPlug, compare ACG against **simple heuristics** (e.g., pick farthest RoIs, lowest point-count RoIs, lowest predicted IoU, etc.). This would show whether the learned gate is necessary or if gains come mostly from a straightforward selection rule.
5. Expand detection beyond a single class (even a limited additional class such as truck) or provide a stronger justification for why single-class completion is the right scope.

---

> ### Author Rebuttal · Authors · 2026-03-30
>
> **We sincerely thank Reviewer Pt1P for the detailed and insightful review and for acknowledging our work's strengths.** We address all concerns and key questions below.
>
> > **Q1. Identity vs. Distribution**: W1
>
> **R1.** Thanks for this insightful comment. The MeanFlow identity is exact and provides the mathematical basis for one-step transport. The stop-gradient objective approximates this identity in practice. This parallels consistency models, which also lack formal distributional guarantees.
>
> Our experimental results directly validate the quality of the learned transport:
>
> - 3DMF achieves competitive CD at 1NFE and the best CD across all completion categories at 5NFE.
> - Table 14a confirms that activating average-velocity learning (r≠t) is essential for 1NFE, verifying that the identity drives the one-step capability.
>
> ---
>
> > **Q2. Endpoint Extrapolation**: W2, W4, KQ2
>
> **R2.** We analyze the gap between training coverage and endpoint inference:
>
> |t sampler|r sampler|t-r>0.1|t-r>0.5|max(t-r)|1NFE CD|
> |-|-|-|-|-|-|
> |uniform(0,1)|uniform(0,1)|81.0%|25.0%|1.000|3.525|
> |lognorm(-0.4,1)|lognorm(-0.4,1)|73.7%|8.5%|0.961|2.471|
> |**beta(2,2)†**|**uniform(0.65,t)**|**48.4%**|**0.0%**|**0.350**|**1.224**|
> |beta(2,2)†|uniform(0.5,t)|61.1%|0.0%|0.500|1.318|
>
> Our default sampler caps max(t−r) at 0.35 with zero endpoint-like pairs. Enlarging step coverage consistently degrades 1NFE, as large steps at high-noise states produce noisy velocity targets that destabilize training.
>
> Despite this, 1NFE still produces reasonable outputs because:
>
> - The average velocity integrates instantaneous velocity over [r,t], and the MeanFlow identity generalizes from trained intervals to [0,1].
> - The shape consistency loss constrains global geometry beyond training coverage.
>
> For multi-step degradation beyond ~5NFE: quality is not inherently tied to step count. As NFE increases, sub-intervals deviate from the trained distribution and errors compound.
>
> ---
>
> > **Q3. JVP & Presentation**: W3, PW1, PW2, PW3
>
> **R3.** Thanks for the suggestions. We will address all these presentation issues in the camera-ready version.
>
> ---
>
> > **Q4. Multi-Step Degradation**: W5
>
> **R4.** Please see our response to Reviewer *fEbd R1*.
>
> ---
>
> > **Q5. nuScenes Metric Scope**: SW1
>
> **R5.** Please see our response to Reviewer *LPjf R2*.
>
> ---
>
> > **Q6. Car-Only Completion**: SW2, EW5
>
> **R6.** Please see our response to Reviewer *fEbd R3*.
>
> ---
>
> > **Q7. Per-Category Training**: SW3
>
> **R7.** We have added multi-category training results, as presented in our response to Reviewer *fEbd R1* and *aHpm R1*.
>
> ---
>
> > **Q8. Adaptation Novelty**: OW1
>
> **R8.** Thanks for the comment. While the core identity is derived from MeanFlow, directly applying it to 3D point clouds requires addressing two challenges:
>
> - **Permutation equivariance**: The original identity is order-sensitive. We reformulate it in a set-based form. Table 16 shows breaking equivariance causes ΔCD from 0.001 to 0.364.
>
> - **Shape-level consistency**: Local velocity targets alone do not regulate global shape. We add a shape-level constraint. Table 6 shows removing it substantially degrades both CD and EMD.
>
> ---
>
> > **Q9. ACG vs. Heuristics**: OW2, OW3, EW4, KQ4
>
> **R9.** We compare simple heuristics (top-K per frame) with ACG and oracle on CID validation, reporting mean ΔIoU:
>
> ||Random|Range (Farthest)|Points (Fewest)|Pred. IoU (Lowest)|Det. Score (Lowest)|Heatmap (Highest)|ACG (Ours)|Oracle (Max ΔIoU)|
> |-|-|-|-|-|-|-|-|-|
> |Mean ΔIoU (×10³)|-123.1|-41.6|-66.9|-5.3|-28.3|-6.7|**+1.2**|+7.7|
>
> All single-attribute heuristics yield negative ΔIoU. ACG is the only method achieving positive gain by combining complementary signals.
>
> We further measure Permutation Feature Importance *I%(f)* to quantify each input's contribution to ACG:
>
> ||Range|Points|Pred. IoU|Det. Score|Heatmap|Proposal Emb.|
> |-|-|-|-|-|-|-|
> |I%(f)|22.0%|34.6%|111.1%|3.7%|61.9%|70.6%|
>
> The top-3 contributors are Pred. IoU, Proposal Emb., and Heatmap, none of which are captured by simple heuristics.
>
> ---
>
> > **Q10. 1NFE Quality Gap**: EW1
>
> **R10.** Please see our response to Reviewer *LPjf R1*.
>
> ---
>
> > **Q11. Timing Fairness**: EW2, KQ3
>
> **R11.** Thanks for this constructive suggestion. The sampling times in our paper are cross-referenced from prior works. To ensure fairness, we re-run all baselines on the same hardware (single A800, batch=1, N=2048).
>
> |Model|PointFlow|PVD|DPF-Net|PSF|3DMF-1NFE|3DMF-5NFE|
> |-|-|-|-|-|-|-|
> |Time (s)|0.184|20.37|0.227|0.061|0.005|0.013|
>
> 3DMF-1NFE remains the fastest method, **12× faster** than PSF. We will update these numbers in the camera-ready version.
>
> ---
>
> > **Q12. Sparsity Sweep**: EW3
>
> **R12.** Please see our response to Reviewer *aHpm R5*.
>
> ---
>
> > **Q13. Detection Training Data**: KQ1
>
> **R13.** Yes, the 3DMF model is trained purely on ShapeNet Car. Real LiDAR scans lack ground-truth complete shapes, so supervised fine-tuning is not applicable.

---

> > ### Author Rebuttal · Reviewer_Pt1P · 2026-04-02
> >
> > Thank you for the detailed rebuttal. Several of my earlier questions are now much clearer.
> >
> > Points that are largely addressed
> > - Endpoint extrapolation: The new sampling analysis is helpful. It makes clear that the default training only covers short intervals (max $t-r=0.35$), and that explicitly enlarging interval coverage hurts 1NFE. This clarifies what the one-step model actually sees during training.
> > - Timing fairness: Re-running baselines on the same hardware/settings addresses this concern well.
> > - ACG vs. heuristics: This is convincingly answered. ACG is the only method with positive mean $\Delta$ IoU, so the learned selector appears justified.
> > - Detection training setup: It is now clear that the detection-time completion model is trained only on ShapeNet Car.
> >
> > Points that remain only partially addressed
> > - My main theory concern remains. The rebuttal still relies on the exact MeanFlow identity plus empirical evidence, but it does not explain under what conditions the learned stop-gradient objective recovers the intended data-to-noise transport, especially for true 1-step inference.
> > - The central 1NFE claim is still mixed. The new endpoint analysis is useful, but the paper results still show that the strongest completion quality often comes from 5NFE rather than 1NFE.
> > - The explanation for degradation beyond about 5 NFE is still qualitative.
> > - The broader practical scope remains limited. The concerns about official nuScenes metrics, car-only detection, multi-category/generalization, and some requested experiments such as sparsity analysis are mostly deferred to other responses or not shown here in enough detail.
> > - The JVP/presentation issues are acknowledged, but only postponed to the camera-ready version. The concern about possible density artifacts from the Chamfer regularizer is also still not directly analyzed.
> >
> > Overall judgment
> > The rebuttal improves my assessment slightly, but not enough to change my recommendation. I would keep weak reject and would not recommend acceptance in the current form. The paper has clear strengths, especially in speed, the permutation-equivariant set formulation, and the PointPlug integration. However, the remaining gaps in theoretical support, the still uneven evidence for the true 1-step claim, and the limited downstream validation under standard evaluation settings are still significant.

---

> > > ### Author Response · Authors · 2026-04-07
> > >
> > > We appreciate your careful re-evaluation. Below we address each remaining point and hope to fully resolve your concerns.
> > >
> > > >1. "...what conditions the learned stop-gradient objective recovers the intended data-to-noise transport..."
> > >
> > > The MeanFlow identity is exact. The stop-gradient objective reaches zero if and only if $\bar{v}_\theta$ satisfies this identity across all intervals, so minimizing the training loss directly drives the model toward the fixed point where the learned transport is provably correct. **The remaining question is whether practical optimization converges to this fixed point.**
> > >
> > > We provide converging experimental evidence from multiple angles:
> > >
> > > - **Correct velocity field**: Strong completion quality and competitive generation confirm accurate learned transport.
> > > - **Extrapolation beyond training range**: Training caps max(t−r) at 0.35, yet 1NFE over full [0,1] works well, showing self-consistency enables reliable extrapolation beyond the training range.
> > > - **Cross-interval coherence**: Multi-step inference produces high-quality results, confirming consistency across interval lengths.
> > >
> > > A formal convergence proof remains open, shared by Consistency Models [1], Shortcut Models [2], and other few-step methods. We hope to contribute towards addressing this gap in future work.
> > >
> > > ---
> > >
> > > >2. "The central 1NFE claim is still mixed..."
> > >
> > > 1NFE may appear "mixed" on quality alone, but quality is not the full claim. At **5 ms per shape** (8× faster than PSF, 5,000× faster than PVD), 1NFE achieves second-best CD on Car completion. As stated in our paper, the contribution of 1NFE is "*one-step sampling with an order-of-magnitude speedup while maintaining competitive fidelity.*"
> > >
> > > That 5NFE further improves quality is a strength, not a contradiction: 1NFE targets hard real-time budgets, while 5NFE offers higher accuracy at a fraction of baseline cost.
> > >
> > > ---
> > >
> > > >3. "The explanation for degradation beyond about 5 NFE is still qualitative."
> > >
> > > We provide a more concrete, mechanistic explanation below. 3DMF's average velocity is trained over intervals biased toward large time spans. At 1–5 NFE, each step covers a large interval matching this training distribution. At higher NFE, the many small steps create intervals rarely seen during training, leading to accumulated approximation error. A similar pattern is observed in other few-step models [3, 4]. We plan to include a more detailed analysis in future work.
> > >
> > > ---
> > >
> > > >4. "...***official nuScenes metrics, car-only detection, multi-category/generalization ... sparsity analysis*** are mostly deferred to other responses or not shown here..."
> > >
> > > Due to character limits, detailed results are cross-referenced to other reviewer responses. Full experimental tables are also provided at [this anonymous link](https://anonymous.4open.science/r/ICML-Rebuttal-19905-Pt1P/README.md). We consolidate the key results for each concern below.
> > >
> > > >(a) Official nuScenes Metrics
> > >
> > > PointPlug yields consistent improvements across all range bins and detectors on both nuScenes and KITTI. Within 0–50 m, completion already improves detection, while at 50 m+, completion provides the **largest relative gains where detection is hardest**.
> > >
> > > >(b) Car-Only Detection
> > >
> > > We train a separate completion model for **Bus** and evaluate on FSHNet (Table C).
> > >
> > > Adding Bus completion yields **+0.3 Bus AP**, **+0.1 mAP**, confirming PointPlug extends beyond a single category.
> > >
> > > >(c) Generalization
> > >
> > > To evaluate generalization, we conduct two additional experiments.
> > >
> > > Joint training on all 55 ShapeNet categories (Table A) shows competitive quality.
> > >
> > > On **ShapeNet-55** (Table E), 3DMF-5NFE achieves the **best F1**, confirming strong multi-category quality.
> > >
> > > >(d) Sparsity Analysis
> > >
> > > Without retraining, 5NFE CD only rises from 0.975 to 1.412 at 20 observed points (Table F), confirming robustness across input sparsity.
> > >
> > > ---
> > >
> > > >5. "The JVP/presentation..."
> > >
> > > We appreciate these suggestions, and outline our camera-ready revision plan: (1) unify notation with a consolidated table; (2) add training pseudocode clarifying JVP computation; (3) move essential PointPlug details from appendix to main text.
> > >
> > > ---
> > >
> > > >6. "...density artifacts from the Chamfer regularizer..."
> > >
> > > Our reported metrics already provide quantitative evidence against significant density artifacts.
> > >
> > > EMD enforces bijective matching, naturally penalizing non-uniform density: if points clustered in certain regions while leaving others sparse, the unmatched regions would incur high EMD. Our EMD remains **competitive across all categories**.
> > >
> > > Additionally, on ShapeNet-55, 3DMF-5NFE achieves the **highest F1** among all methods, where high recall specifically rules out regional clustering.
> > >
> > > [1] Song et al., Consistency Models, ICML 2023.
> > >
> > > [2] Frans et al., One Step Diffusion via Shortcut Models, ICLR 2025.
> > >
> > > [3] Sabour et al., Align Your Flow, NeurIPS 2025.
> > >
> > > [4] Dao et al., Self-Corrected Flow Distillation, AAAI 2025.

---

### Official Review · Reviewer_LPjf · 2026-03-12

**Soundness:** 3
**Presentation:** 3
**Significance:** 3
**Originality:** 2
**Overall Recommendation:** 4
**Confidence:** 4

**Summary:**

This paper proposes 3D MeanFlow (3DMF), a teacher-free approach for one-step (and few-step) point cloud completion and unconditional generation, adapting the MeanFlow average-velocity identity to permutation-equivariant point sets. The method trains an interval-averaged velocity field via an instantaneous-average consistency objective and adds a shape-level Chamfer constraint to stabilize global geometry. The paper also introduces PointPlug, a module that applies one-step completion on selected detector RoIs to refine 3D detection outputs, with an adaptive completion gate. Experiments report completion/generation trade-offs on ShapeNet categories and detection improvements on nuScenes and KITTI.

**Compliance With Llm Reviewing Policy:**

Affirmed.

**Key Questions For Authors:**

Tables 3 and 4 compare detector vs detector+PointPlug, but PointPlug includes multiple interacting components (ACG ranking, box refitting, score fusion, and the completion model). To attribute improvements to completion rather than involving box fitting heuristics or score calibration, comparisons with following configurations are highly encouraged:
1. Fit boxes using visible RoI points only (no completed points), keeping the rest identical,
2. Swap 3DMF with another completion model (even a slower one) for an accuracy-controlled comparison.

**Limitations:**

Yes

**Strengths And Weaknesses:**

Strengths:
1. Clear adaptation of MeanFlow-style training to point sets with permutation constraints. The paper explicitly states permutation equivariance requirements and uses them to justify a set-based MeanFlow identity, which is the right kind of formalism for unordered point clouds.
2. Practical one-step sampling interface and training schematic. Figure 3 does a good job separating the stop-gradient branch constructing  (via JVP) from the forward completion update, making the training mechanics more interpretable than many “few-step” papers that bury these details.
3. Speed is genuinely compelling in the reported protocol. In Table 1, 3DMF-1NFE reports 0.005s per shape, substantially faster than PSF’s 0.04s (and far faster than diffusion baselines), which is relevant for real-time completion. Figure 1’s bubble plots succinctly visualize the speed-quality frontier for both completion and generation.

Weakness:
1. In Table 1, 3DMF-1NFE has substantially worse CD than all listed baselines on Airplane and Chair. Only Car improves noticeably (1.224 vs 1.384/1.396/1.803). This is an important gap because the paper repeatedly emphasizes “competitive fidelity” in one step.
2. In Table 2, for Airplane and Car, 3DMF-1NFE has higher 1-NNA than many baselines (values farther from 50 are worse per the caption), e.g., Airplane CD-based 1-NNA is 76.19 (worse than PSF 71.11, PVD 73.82), and Chair is not best either.
3. In Table 3, NDS improves by +0.1 to +0.2 and mAP by +0.4 at most. The biggest relative gains are in the 50m+ bin, but the paper also notes that official nuScenes metrics exclude 50m+. This limits the impact on downstream tasks.

---

> ### Author Rebuttal · Authors · 2026-03-30
>
> **We sincerely thank Reviewer LPjf for the constructive comments and actionable suggestions.** We are glad the reviewer finds our formulation and training design well-motivated. We address the remaining concerns and questions below.
>
> > **Q1. Speed–Quality Trade-off of 1NFE**
> >
> > - *Weakness 1*: "In Table 1, 3DMF-1NFE has substantially worse CD than all listed baselines on Airplane and Chair. Only Car improves noticeably..."
> >
> > - *Weakness 2*: "In Table 2, ...3DMF-1NFE has higher 1-NNA than many baselines..."
>
> **R1.** Thanks for the detailed numerical analysis. Both points question whether 1NFE quality is truly competitive, and we argue it is once the speed context is considered.
>
> 3DMF-1NFE is **an order of magnitude faster** than all baselines in Tables 1 and 2:
>
> - **5 ms per shape**, **8× faster** than PSF (0.04 s) and over **5,000× faster** than PVD (29.9 s)
> - Despite this extreme compression to a single forward pass, 1NFE already delivers:
>   - **Car completion**: second-best CD, behind only our own 3DMF-5NFE
>   - **Car generation**: competitive 1-NNA, on par with the strongest baselines
>
> This confirms that even at the one-step limit, the learned average-velocity transport produces meaningful outputs.
>
> When modest additional latency is acceptable, 3DMF-5NFE further delivers:
>
> - **Best CD on all three completion categories** (Airplane, Chair, Car)
> - **Best 1-NNA on Car generation**
>
> The two settings serve complementary roles: **1NFE targets hard real-time budgets** where no existing method can operate, while **5NFE offers state-of-the-art accuracy** at a fraction of competing methods' cost.
>
> ------
>
>
>
> > **Q2. Limited Impact on Downstream Detection**
> >
> > - *Weakness 3*: "In Table 3, NDS improves by +0.1 to +0.2 and mAP by +0.4 at most. The biggest relative gains are in the 50m+ bin, but...official nuScenes metrics exclude 50m+."
>
> **R2.** Thanks for the comment. We note that PointPlug yields **consistent improvements across all range bins and all evaluated detectors** on both nuScenes and KITTI, confirming that point cloud completion is a meaningful addition to real-world 3D detection pipelines.
>
> The range-dependent gain pattern has a natural explanation:
>
> - **0–50 m**: LiDAR points are already relatively dense, so detectors already perform well and the room for completion to help is naturally limited.
> - **50 m+**: Point clouds become increasingly sparse and object geometry is under-determined. Completion fills in missing structure, providing the **largest relative gains precisely where detection is hardest**.
>
> The fact that PointPlug delivers its strongest improvements in these challenging regimes highlights the **robustness** of our completion model under real LiDAR sparsity conditions.
>
> ------
>
>
>
> > **Q3. Ablation of PointPlug Components**
> >
> > - *Key Question 1*: "Fit boxes using visible RoI points only (no completed points), keeping the rest identical."
> > - *Key Question 2*: "Swap 3DMF with another completion model (even a slower one) for an accuracy-controlled comparison."
>
> **R3.** Thanks for the suggestions. Following the reviewer's advice, we conduct a component-wise ablation of PointPlug on FSHNet and report results below.
>
> *Table G. Ablation of PointPlug components on FSHNet (nuScenes val).*
>
> | Completion Model | ACG  | Box Revise | Score Revise | NDS  | mAP  | Time (ms) |
> | :--------------: | :--: | :--------: | :----------: | :--: | :--: | :-------: |
> |   - (baseline)   |      |            |              | 71.7 | 68.1 |    123    |
> |       3DMF       |      |     ✓      |      ✓       | 71.2 | 67.4 |    128    |
> |       3DMF       |  ✓   |            |      ✓       | 71.5 | 67.8 |    129    |
> |       3DMF       |  ✓   |     ✓      |              | 71.7 | 68.2 |    129    |
> |        -         |  ✓   |     ✓      |      ✓       | 71.7 | 68.0 |    124    |
> |       PSF        |  ✓   |     ✓      |      ✓       | 71.7 | 68.3 |    188    |
> |   3DMF (ours)    |  ✓   |     ✓      |      ✓       | 71.8 | 68.5 |    129    |
>
>
> All components of PointPlug are essential to the final result:
>
> - **ACG**: Removing ACG causes the largest degradation. Without selective gating, completion is applied to randomly chosen proposals, which, as shown in Figure 7, are more likely to incur negative BEV-IoU changes than positive ones.
> - **Box Revise & Score Revise**: These two components act as complementary parts, removing either one leads to a noticeable drop, confirming that both geometric refinement and score calibration are necessary for the full gain.
> - **Completion model swap**: Replacing 3DMF with PSF still yields a positive improvement over the baseline, validating that the PointPlug framework generalizes across completion backbones. However, this comes at a **higher latency**, highlighting 3DMF's one-step efficiency advantage.

---

> > ### Author Rebuttal · Reviewer_LPjf · 2026-04-03
> >
> > Thank you for the thorough rebuttal. I appreciate the detailed clarifications regarding the speed–quality trade-off of 1NFE, the range-dependent gains on downstream detection, and the component-wise ablation of PointPlug. All my concerns have been adequately addressed. Therefore, I maintain my positive score and recommend acceptance.

---

> > > ### Author Response · Authors · 2026-04-07
> > >
> > > Thank you for the encouraging response and for recommending acceptance. We are delighted that our clarifications have fully addressed your concerns. We sincerely appreciate the rigorous and constructive feedback throughout the review process, which has meaningfully improved our work.

---

### Official Review · Reviewer_aHpm · 2026-03-12

**Soundness:** 3
**Presentation:** 3
**Significance:** 3
**Originality:** 3
**Overall Recommendation:** 4
**Confidence:** 3

**Summary:**

This paper proposes 3D MeanFlow (3DMF), a distillation-free one-step framework for point cloud completion and generation based on an average-velocity transport formulation. The method predicts an average flow to enable single-step sampling, aiming to reduce the inference cost of diffusion-style generative models. The paper also introduces PointPlug, which integrates the completion model into 3D object detectors to refine point clouds within proposals. Experiments on ShapeNet, KITTI, and nuScenes demonstrate fast sampling and improvements in downstream 3D detection performance.

**Compliance With Llm Reviewing Policy:**

Affirmed.

**Final Justification:**

The rebuttal addresses my main concerns with additional baselines and analyses, strengthening the paper. While broader validation beyond ShapeNet and clearer explanation of the NFE trade-off would be helpful, I maintain my positive score unchanged.

**Key Questions For Authors:**

1. Quality–speed trade-off of one-step MeanFlow.
   The proposed method focuses on single-step sampling for efficiency. Could the authors provide more analysis on the trade-off between generation quality and the number of sampling steps (e.g., 1-step vs. a few-step MeanFlow)? Understanding how performance changes with additional steps would help clarify whether the proposed formulation is fundamentally limited to one-step inference or can also benefit from multi-step refinement.

2. Failure cases and robustness.
   Could the authors provide more analysis on failure cases or challenging scenarios for the proposed model (e.g., highly incomplete point clouds or complex object geometries)?

**Limitations:**

yes

**Strengths And Weaknesses:**

### Strengths
- Clear writing and strong motivation.
The paper is clearly written and the motivation for reducing the inference cost of point cloud generation is well explained.

- Efficient generative formulation.
The MeanFlow formulation enables single-step sampling, significantly reducing inference cost compared to diffusion or multi-step flow models.

- Connection to downstream tasks.
The proposed PointPlug module demonstrates that point cloud completion can improve real-world 3D detection performance.

- Conceptually simple design.
The method is straightforward and can be integrated with existing point-based architectures.

### Weaknesses

- Incomplete baseline comparisons. The completion experiments omit several important baselines such as [1, 2, 3], which are widely used in the completion literature but not included in the quantitative comparison.

- Simplified generation setting. The generation experiments are limited to ShapeNet object generation, which is a relatively simple closed-set benchmark and does not reflect more realistic 3D environments such as scene-level or LiDAR point cloud generation [4, 5].

- Limited qualitative visualization. The qualitative results are somewhat limited. For example, the completion visualization only includes three object categories. Providing more diverse qualitative examples would help better demonstrate the generalization capability of the method.

[1] PCN: Point Completion Network

[2] SeedFormer: Patch Seeds based Point Cloud Completion with Upsample Transformer

[3] EINet: Explicit Inference Network for Point Cloud Completion

[4] RangeLDM: Fast Realistic LiDAR Point Cloud Generation

[5] TIGER: Time-Varying Denoising Model for 3D Point Cloud Generation”

---

> ### Author Rebuttal · Authors · 2026-03-30
>
> **We sincerely thank Reviewer aHpm for the thorough evaluation, concrete suggestions, and insightful questions.**  We are encouraged that the reviewer appreciates the efficient single-step sampling, the practical value of PointPlug for downstream detection, and the simplicity of the overall design. We address the remaining concerns and questions below.
>
> > **Q1. Incomplete Baseline Comparisons**
> >
> > - *Weakness 1*: "The completion experiments omit several important baselines such as [1, 2, 3]...not included in the quantitative comparison."
>
> **R1.** Thanks for the suggestion. We train 3DMF on **ShapeNet-55** and compare with PCN [1], SeedFormer [2], and EINet [3].
>
> *Table E. Multi-category completion on ShapeNet-55.*
>
> |Model|Table|Chair|Airplane|Car|Sofa|CD-S|CD-M|CD-H|CD-Avg|F1|
> |:-:|:-:|:-:|:-:|:-:|:-:|:-:|:-:|:-:|:-:|:-:|
> |PCN|2.13|2.29|1.02|1.85|2.06|1.94|1.96|4.08|2.66|0.133|
> |SeedFormer|0.72|0.81|**0.40**|0.89|0.69|0.50|0.77|1.49|0.92|0.472|
> |EINet|**0.66**|0.79|0.41|0.84|0.71|0.49|**0.75**|**1.46**|**0.90**|0.432|
> |3DMF-1NFE|0.77|0.83|0.48|0.87|0.69|0.54|0.82|1.60|0.98|0.391|
> |3DMF-5NFE|0.70|**0.78**|0.44|**0.82**|**0.66**|**0.47**|**0.75**|1.50|0.91|**0.480**|
>
> 3DMF-5NFE achieves the **best F1** and ranks first on **5/8 CD metrics**.
>
> ------
>
> > **Q2. Generation Beyond ShapeNet**
> >
> > - *Weakness 2*: "The generation experiments are limited to ShapeNet...does not reflect more realistic 3D environments such as scene-level or LiDAR point cloud generation [4, 5]."
>
> **R2.** Thanks for the comment. Our generation scope is **object-level**, which differs from scene-level LiDAR point cloud generation (e.g., RangeLDM [4]) in data representation and scale.
>
> For real-world LiDAR point clouds, our PointPlug experiments apply 3DMF completion on nuScenes and KITTI. Since real LiDAR scans lack ground-truth complete shapes, the consistent detection improvements indirectly validate that 3DMF generalizes effectively beyond ShapeNet.
>
> ------
>
> > **Q3. Limited Qualitative Visualization**
> >
> > - *Weakness 3*: "The qualitative results are somewhat limited...only includes three object categories...more diverse qualitative examples would help better demonstrate the generalization capability."
>
> **R3.** Thank you for your feedback. We provide additional qualitative results beyond the three main categories in Appendix: **Figure 9** shows completion comparisons on more ShapeNet categories, and **Figure 11** shows unconditional generation results across diverse categories.
>
> ------
>
> > **Q4. Quality–Speed Trade-off and Multi-Step Refinement**
> >
> > - *Key Question 1*: "...more analysis on the trade-off between generation quality and the number of sampling steps...whether the proposed formulation is fundamentally limited to one-step inference or can also benefit from multi-step refinement."
>
> **R4.** Thanks for the question. We provide a full ablation of sampling steps in Appendix E (Table 15), covering both quality and sampling time. The 3DMF formulation naturally supports any number of sampling steps (Section 3.4) and is not limited to one-step inference.
>
> Regarding the trade-off: 1NFE is designed for **extreme efficiency**, completing a shape in only 5 ms while maintaining competitive quality. 5NFE reaches the best overall quality at just 13 ms, striking the best balance between speed and quality.
>
> ------
>
> > **Q5. Failure Cases and Robustness**
> >
> > - *Key Question 2*: "...failure cases or challenging scenarios...highly incomplete point clouds or complex object geometries?"
>
> **R5.** Thanks for the question. To evaluate robustness, we test our model on different input sizes without retraining:
>
> *Table F. Completion under varying input sparsity on ShapeNet Car.*
>
> |Observed Points|1NFE CD↓|1NFE EMD↓|5NFE CD↓|5NFE EMD↓|
> |:-:|:-:|:-:|:-:|:-:|
> |20|1.567|2.099|1.412|1.972|
> |50|1.285|1.981|1.107|1.770|
> |100|1.237|1.904|0.981|1.765|
> |200 (default)|1.224|1.896|0.975|1.752|
>
> Performance degrades gracefully: even at **20 observed points**, 5NFE CD only rises from 0.975 to 1.412, confirming robustness across a wide range of input sparsity. Following PVD, partial point clouds are obtained from **20 random views** per shape via hidden point removal, so the model is already exposed to diverse occlusion patterns during training.
>
> We visualize representative failure cases at **20 input points** in [Fig. A](https://anonymous.4open.science/r/ICML-Rebuttal-19905/rebuttal.png). Typical failures include:
>
> - **fine structures** (e.g., airplane tails in Cases 1–2).
> - **missing dominant structures** (e.g., airplane wings in Case 1, chair styles in Cases 3–4).
>
> We will include a detailed discussion in the camera-ready version.
>
>
>
> [1] Yuan et al., PCN, 3DV 2018.
>
> [2] Zhou et al., SeedFormer, ECCV 2022.
>
> [3] Cai et al., EINet, ECCV 2024.
>
> [4] Hu et al., RangeLDM, ECCV 2024.

---

> > ### Author Rebuttal · Reviewer_aHpm · 2026-04-02
> >
> > Thank you for the detailed rebuttal and the additional experiments, which address several of my original concerns. In particular, the newly added completion baselines and robustness analysis are helpful. I still hope to see generation results beyond ShapeNet to better establish the generality of the proposed generative framework, since the current evidence remains largely object-level and closed-set. I also find the quality-speed trade-off results interesting but somewhat non-intuitive, as performance appears to peak around 5 NFEs and can degrade with more steps (e.g., 50 NFEs underperforming 5 NFEs), which would benefit from further explanation. Overall, the rebuttal resolves my main questions and strengthens the paper, although I still see some room for broader validation, so I maintain my positive scores.

---

> > > ### Author Response · Authors · 2026-04-07
> > >
> > > Thank you sincerely for the thoughtful acknowledgement and for taking the time to carefully review our additional experiments. Your suggestions on additional baselines and robustness analysis have directly improved our work, and we are encouraged by your continued positive assessment.
> > >
> > > We briefly address the two remaining points below and hope the following clarifications fully resolve your concerns.
> > >
> > > > **1. Evidence Beyond ShapeNet**
> > > >
> > > > - "I still hope to see generation results beyond ShapeNet to better establish the generality of the proposed generative framework, since the current evidence remains largely object-level and closed-set."
> > >
> > > We are glad to present additional evidence beyond closed-set settings. We present completion results on real-world LiDAR point clouds from **nuScenes** and **KITTI**. As shown in [Fig. B](https://anonymous.4open.science/r/ICML-Rebuttal-19905-aHpm/rebuttal_lidarCompletion.png), our 3DMF-1NFE model produces convincing completions on these samples, which exhibit substantially different point distributions and sparsity patterns compared to synthetic ShapeNet data.
> > >
> > > Since real-world LiDAR scans do not come with ground-truth complete shapes, we offer these as qualitative evidence that 3DMF generalizes effectively beyond closed-set settings.
> > >
> > > Regarding the object-level setting, 3DMF follows the standard evaluation protocol in point cloud completion and generation, where the compared methods (PVD, PSF, etc.) operate on individual objects. Our method operates in a fixed-dimensional space $\mathbb{R}^{N \times 3}$, which cannot directly accommodate scene-level point clouds with variable and significantly larger point counts.
> > >
> > > ------
> > >
> > >
> > >
> > > > **2. Multi-Step Degradation**
> > > >
> > > > - "I also find the quality-speed trade-off results interesting but somewhat non-intuitive, as performance appears to peak around 5 NFEs and can degrade with more steps (e.g., 50 NFEs underperforming 5 NFEs), which would benefit from further explanation."
> > >
> > > Thank you for the keen observation. We provide our explanation below.
> > >
> > > The velocity field in 3DMF is trained to approximate the average velocity over (t, r) intervals that are **biased toward large time spans**. When using a few inference steps (e.g., 1 or 5 NFEs), each step covers a large interval that aligns well with this training distribution. However, subdividing into many small steps (e.g., 50 NFEs) creates intervals the model has rarely seen during training, which can lead to **accumulated approximation error**.
> > >
> > > A similar phenomenon has been observed in other few-step generative models [5, 6], suggesting this may be a shared characteristic of models trained for few-step generation. We will include a more detailed analysis of such multi-step degradation in future work.
> > >
> > > [5] Sabour et al., Align Your Flow: Scaling Continuous-Time Flow Map Distillation, NeurIPS 2025.
> > >
> > > [6] Dao et al., Self-Corrected Flow Distillation for Consistent One-Step and Few-Step Image Generation, AAAI 2025.

---

### Official Review · Reviewer_fEbd · 2026-03-14

**Soundness:** 3
**Presentation:** 3
**Significance:** 3
**Originality:** 3
**Overall Recommendation:** 4
**Confidence:** 3

**Summary:**

The paper introduces 3D MeanFlow (3DMF), which extends the mean-flow one-step sampling method to unordered 3D point sets. The objective is to learn an average-velocity field in 3D space that transports randomly sampled points to a clean point-cloud distribution in a single sampling step. The authors propose: (1) a formulation of average-velocity transport for unordered, permutation-invariant 3D point clouds; (2) an additional shape-level constraint that enables stable one-step generation for point clouds; and (3) the application of the method as a lightweight point-cloud completion module to improve car detection in outdoor scenes. Experimental results show that the proposed method is up to 8× faster than state-of-the-art fast point-cloud generation methods while maintaining strong performance.

**Compliance With Llm Reviewing Policy:**

Affirmed.

**Final Justification:**

The authors adressed most of my concerns therefore I maintain my positive score while increase my confidence rating.

**Key Questions For Authors:**

- How does 3DMF perform when trained on a large-scale, multi-category dataset (like the full ShapeNet) rather than category-specific models? Does the shape-level consistency term still prevent drift effectively?
- What is the key component the make the proposed method faster than PSF?

**Limitations:**

yes

**Strengths And Weaknesses:**

Strength:
- The paper is well written.
- Fast point cloud completion is an important topic with direct benefits for robotics and autonomous driving. It is valuable that the authors demonstrate the application of fast point cloud completion to support object detection in outdoor scenes.
- The proposed shape-level constraint is a reasonable approach to improve single-step generation quality. The ablation study clearly demonstrates its effectiveness.
- The formal formulations in the paper strengthen the soundness of adopting the mean flow framework for the 3D point cloud domain.

Weakness:
- It is unclear whether the model is trained separately for each category or jointly across all ShapeNet categories.
- Only a few categories are evaluated: airplane, chair, and car in ShapeNet. For the KITTI dataset, the detection experiment only tests the car category.
- PSF is also a fast single-step flow-based point cloud generation method that models a straight transport field in 3D space. It outperforms the proposed method in both CD and EMD for the airplane category while showing comparable performance in other categories (Table 2). The speed advantage of the proposed method may stem from differences in the denoising backbone rather than the adoption of mean flow. This point deserves further discussion.

---

> ### Author Rebuttal · Authors · 2026-03-29
>
> **We sincerely thank Reviewer fEbd for the encouraging assessment and the constructive suggestions.**  We appreciate the recognition of the practical value of fast completion for detection, the effectiveness of the shape-level constraint, and the soundness of our formal formulations. We address the remaining concerns and questions below.
>
> > **Q1. Multi-category Completion Evaluation**
> > - *Weakness 1:* "It is unclear whether the model is **trained separately for each category or jointly**..."
> > - *Key Question 1:* "How does 3DMF perform when trained on a large-scale, multi-category dataset...? Does the shape-level consistency term still prevent drift...?"
>
> **R1.** Thanks for the question. All results in our paper use **per-category training**, consistent with baselines (PSF, PVD, etc.).
>
> To evaluate the impact of joint training, we train a single 3DMF model on all 55 ShapeNet categories and compare with our per-category models.
>
> *Table A. Per-category vs. joint training on ShapeNet.*
>
> |Model|Airplane CD↓|Airplane EMD↓|Chair CD↓|Chair EMD↓|Car CD↓|Car EMD↓|
> |-|-|-|-|-|-|-|
> |3DMF-1NFE|0.639|1.167|2.933|3.541|1.224|1.896|
> |3DMF-5NFE|0.369|1.121|2.138|3.188|0.975|1.752|
> |3DMF-1NFE (joint)|0.725|1.324|3.573|4.170|1.418|2.163|
> |3DMF-5NFE (joint)|0.412|1.279|2.559|3.762|1.191|2.013|
>
> 3DMF maintains competitive completion quality under joint training across all three categories, confirming that the average-velocity formulation generalizes well to multi-category settings.
>
> For the shape-level consistency term: Table 6 shows both CD and EMD are optimal at λ=2.0. Notably, EMD is a bijection-based metric sensitive to density distribution, so its minimum at λ=2.0 indicates that stronger shape regularization does not introduce density artifacts. The joint training results here further confirm this design remains effective across diverse categories.
>
> ---
>
> > **Q2. Additional Category Evaluation**
> >
> > - *Weakness 2:* "Only a few categories are evaluated..."
>
> **R2.** Thanks for the comment. Most prior methods only report Airplane, Chair, and Car. Since these methods do not provide per-category results on other classes or release pretrained weights, we retrain PointFlow, DPF-Net, and PVD locally on **Sofa** and **Table** and report the results below.
>
> *Table B. Per-category completion on Sofa and Table.*
>
> |Model|Sofa CD↓|Sofa EMD↓|Table CD↓|Table EMD↓|Time (s)|
> |-|-|-|-|-|-|
> |PointFlow|1.901|2.246|2.956|3.273|0.27|
> |DPF-Net|1.890|2.013|3.007|2.918|0.34|
> |PVD|2.145|1.812|3.539|2.624|29.9|
> |3DMF-1NFE|1.872|2.046|3.461|2.866|0.005|
> |3DMF-5NFE|1.486|1.860|2.161|2.778|0.013|
>
> 3DMF achieves competitive quality on both categories while maintaining a significant speed advantage. We also show qualitative results on more categories in Figures 9 and 11.
>
> ---
>
> > **Q3. Multi-class Detection Evaluation**
> >
> > - *Weakness 2:* "...the detection experiment only tests the car category."
>
> **R3.** Thanks for the comment. To extend the detection evaluation beyond Car, we select **Bus** as an additional class since it overlaps with ShapeNet categories (we exclude Motorcycle due to a domain gap: ShapeNet's motorcycle lacks the rider present in nuScenes' Motor). We train a separate completion model and ACG module for Bus and evaluate on FSHNet.
>
> *Table C. Multi-class detection on nuScenes val with PointPlug.*
>
> |Model|NDS|mAP|Car|Truck|Bus|T.L.|C.V.|Ped.|M.T.|Bike|T.C.|B.R.|
> |-|-|-|-|-|-|-|-|-|-|-|-|-|
> |FSHNet|71.7|68.1|88.7|61.4|79.3|47.8|26.3|89.3|76.7|60.5|78.6|72.3|
> |FSHNet-pp (Car)|71.8|68.5|89.3|61.4|79.2|47.8|26.3|89.3|76.7|60.5|78.6|72.4|
> |FSHNet-pp (Car & Bus)|71.8|68.6|89.3|61.5|79.6|47.8|26.3|89.3|76.7|60.5|78.6|72.4|
>
> Adding Bus completion yields **+0.3 Bus AP** and **+0.1 mAP**, confirming that PointPlug successfully generalizes to additional classes.
>
> ---
>
> > **Q4. 3DMF vs. PSF Timing Analysis**
> >
> > - *Weakness 3:* "PSF is also a fast single-step flow-based point cloud generation method... The speed advantage...stem from differences in the denoising backbone..."
> > - *Key Question 2:* "What is the **key component** that makes the proposed method **faster than PSF**?"
>
> **R4.** Thanks for the question. We compare the following configurations on ShapeNet Car completion to isolate the source of the speed gap:
>
> *Table D. Configuration ablation on ShapeNet Car.*
>
> |Model|Backbone|CD↓|EMD↓|Time (ms)|
> |-|-|-|-|-|
> |PSF|PVCNN2|1.388|2.194|61.29|
> |PSF (no distillation)|PVCNN2|1.369|2.081|24249.60|
> |PSF|PointwiseNet|4.120|3.554|4.52|
> |3DMF|PointwiseNet|1.224|1.896|4.93|
>
> PSF relies heavily on distillation, which is the primary means of compressing multi-step inference into a single step. Replacing its PVCNN2 backbone with our lightweight PointwiseNet leads to severe performance degradation, showing that PSF's distillation framework demands a high-capacity backbone.
>
> In contrast, our average-velocity formulation requires no distillation and works well with a lightweight backbone, which is the fundamental reason behind the speed advantage.

---

> > ### Author Rebuttal · Reviewer_fEbd · 2026-04-03
> >
> > Thank you for the detailed reponse. I am satisfied with the response and will maintain my score while increasing my confidence rating.

---

> > > ### Author Response · Authors · 2026-04-07
> > >
> > > Thank you for the positive reassessment. We are glad that our response has addressed your concerns. We truly appreciate the constructive feedback you provided throughout the review process, and your suggestions have genuinely helped us strengthen the paper.

---

### Decision · Program_Chairs · 2026-04-30

**Decision:**

Accept (regular)

**Comment:**

The paper received 4 weak accept. The reviewers appreciate the effective designs
of the method that significantly improve the generation speed and its applications
on downstream tasks. All reviewers reach a consensus to accept the paper,
and the AC agrees with that decision and recommends acceptance. The authors
are encouraged to further revise the paper based on the reviewer feedback and
include the necessary clarifications from the rebuttal.